# FUNCTIONAL WASSERSTEIN BRIDGE INFERENCE FOR BAYESIAN DEEP LEARNING

## ABSTRACT

Bayesian deep learning (BDL) is an emerging field that combines the strong function approximation power of deep learning with the uncertainty modeling capabilities of Bayesian methods. In addition to those virtues, however, there are accompanying issues brought by such a combination to the classical parameter-space variational inference, such as the nonmeaningful priors, intricate posteriors, and possible pathologies. In this paper, we propose a new function-space variational inference solution called Functional Wasserstein Bridge Inference (FWBI), which can assign meaningful functional priors and obtain well-behaved posterior. Specifically, we develop a Wasserstein distance-based bridge to avoid the potential pathological behaviors of Kullback–Leibler (KL) divergence between stochastic processes that arise in most existing functional variational inference approaches. The derived functional variational objective is well-defined and proved to be a lower bound of the model evidence. We demonstrate the improved predictive performance and better uncertainty quantification of our FWBI on several tasks compared with various parameter-space and function-space variational methods.

## 1 INTRODUCTION

In past decades, Bayesian deep learning (BDL) approaches (Blundell et al., 2015; Gal, 2016; Wilson & Izmailov, 2020) have shown success in combining the strong predictive performance of deep learning models with the principled uncertainty estimation of Bayesian inference. They have been recognized as an effective and irreplaceable tool for a wide range of tasks, such as the uncertainty formulation in per-pixel semantic segmentation (Kendall & Gal, 2017), risk-sensitive reinforcement learning (Depeweg et al., 2018), and safety-critical medical diagnosis and diabetic detection (Filos et al., 2019; Band et al., 2022).

Even though impressive progress has been made, the application of Bayesian deep learning has not achieved outstanding performance in some tasks compared with their non-Bayesian counterparts (Ovadia et al., 2019; Foong et al., 2019; Farquhar et al., 2020). This phenomenon can be probably attributed to at least two unresolved issues in their parameter-space inference. Firstly, it is difficult to incorporate meaningful prior information about the unknown function into the inference procedure. The widely used independent and identically distributed Gaussian priors for model parameters are not always applicable for that, because the samples of such priors over parameters tend to be horizontally linear and lead to pathologies for deep models (Duvenaud et al., 2014; Matthews et al., 2018; Tran et al., 2020) (we visualize this problem in Appendix A for the self-contained purpose). Moreover, the effects of the given priors on posterior inference and further on the resulting distributions over functions are unclear and hard to control owing to the complex architecture and non-linearity of the models (Ma & Hernández-Lobato, 2021; Fortuin et al., 2021; Wild et al., 2022).

To avoid these issues, there has been increasing attention to performing Bayesian inference in function space instead of parameter space (Ma et al., 2019; Rudner et al., 2020; 2022a). In such an inference framework, the distributions of function mappings defined by models are treated as probability measures in function space induced by the distributions over model parameters, and then the variational objective is defined in terms of the distributions over functions directly. In this situation, one can take advantage of more informative stochastic process priors, such as the classic *Gaussian Processes* (GPs) which can easily encode prior knowledge about function properties (e.g. periodicity and smoothness) through corresponding kernel functions. In order to approximate posterior distri-

butions over functions, existing function-space inference methods (Sun et al., 2019) explicitly build and minimize the divergence or distance between the true posterior and the variational posterior processes and develop a tractable estimate procedure for the functional variational objective.

Like parameter-space variational methods, function-space inference methods mostly use the Kullback–Leibler (KL) divergence as the measure of dissimilarity. However, such KL divergence for distributions over infinite-dimensional functions may be infinite (Burt et al., 2020), which leads to the variational objective being ill-defined. Specifically, as a key to the definition of KL divergence between probability measures, the existence of Radon-Nikodym derivatives between the prior and the variational approximate posterior must satisfy that the former is absolutely continuous with respect to the latter (Matthews et al., 2016; Burt et al., 2020), which may not be satisfied in some situations. For example, the KL divergence between distributions over functions generated from two Bayesian neural networks with different network structures can be infinite (Ma & Hernández-Lobato, 2021).

In this work, we investigate a new functional variational inference method using a Wasserstein bridge as a dissimilarity measure for distributions called *Functional Wasserstein Bridge Inference* (FWBI), which can avoid the limitations of KL divergence for distributions over functions. Our main contributions are as follows:

- We propose a new Bayesian inference framework in function space to avoid the limitations of parameter-space mean-field variational inference, such as the difficulties of defining meaningful priors and uncontrolled pathologies from over-parametrization.
- We propose a variational objective in terms of distributions over functions based on the Wasserstein bridge as the alternative for KL divergence between probability measures. We prove that our objective function is a lower bound of the model evidence and therefore is a well-defined objective for Bayesian inference.
- We evaluate the proposed method by comparing it against competing parameter-space and function-space inference approaches on several tasks to demonstrate its highly predictive performance and reliable uncertainty estimation.

## 2 PRELIMINARIES

Consider a supervised learning task with dataset $\mathcal{D} = \{(x_i, y_i)\}_{i=1}^n = \{\mathbf{X}_\mathcal{D}, \mathbf{Y}_\mathcal{D}\}$, where $x_i \in \mathcal{X} \subseteq \mathbb{R}^d$ are the training inputs and $y_i \in \mathcal{Y} \subseteq \mathbb{R}^c$ denote the corresponding targets. Let $f(\cdot; \mathbf{w}) : \mathcal{X} \to \mathcal{Y}$ be a function mapping defined by an arbitrary machine learning model with model parameters $\mathbf{w}$. For example, $f$ can be the function mapping given by a Bayesian Neural Network (BNN), which is one of the most representative BDL models. BNNs are stochastic neural networks, and their parameters (weights) are multivariate random variables resulting in a random function $f(\cdot; \mathbf{w})$, that is, a stochastic process. When evaluated at finite marginal points $\mathbf{X}$ in the input domain, $f(\mathbf{X}; \mathbf{w})$ turns into a multivariate random variable.

**Parameter-space variational inference** BDL models usually are trained with Bayesian inference by placing prior distributions over model parameters, such as $p_0(\mathbf{w})$ is the prior distribution for random network weights in BNNs. Given the training data $\{\mathbf{X}_\mathcal{D}, \mathbf{Y}_\mathcal{D}\}$ and a proper likelihood $p(\mathbf{Y}_\mathcal{D}|\mathbf{X}_\mathcal{D}, \mathbf{w})$ evaluated with the training set, the posterior of weights then can be inferred as $p(\mathbf{w}|\mathcal{D}) \propto p_0(\mathbf{w})p(\mathbf{Y}_\mathcal{D}|\mathbf{X}_\mathcal{D}, \mathbf{w})$. However, due to the non-linear nature of the function mapping $f$ in terms of random $\mathbf{w}$, the marginal integration required in solving the posterior over weights is intractable for any practical dimension. Variational inference (Wainwright et al., 2008) is one of the most popular approximation approaches to convert the problem of estimating the posterior distribution into a tractable optimization problem. The goal of variational inference is to fit an approximate posterior distribution $q(\mathbf{w}; \boldsymbol{\theta_q})$ parametrized by $\boldsymbol{\theta_q}$ from a tractable variational family by minimizing the KL divergence between it and the true posterior as $\min_{q(\mathbf{w};\boldsymbol{\theta_q})} \mathrm{KL}[q(\mathbf{w}; \boldsymbol{\theta_q})\|p(\mathbf{w}|\mathcal{D})]$, which is equivalent to maximizing the evidence lower bound (ELBO) as follows:

$$\mathcal{L}_{q(\mathbf{w};\boldsymbol{\theta_q})} := \mathbb{E}_{q(\mathbf{w};\boldsymbol{\theta_q})}\left[\log p(\mathbf{Y}_\mathcal{D} \mid \mathbf{X}_\mathcal{D}, \mathbf{w}; \boldsymbol{\theta_q})\right] - \mathrm{KL}[q(\mathbf{w}; \boldsymbol{\theta_q})\|p_0(\mathbf{w})]. \qquad (1)$$

**Function-space variational inference** The core idea of function-space variational inference is to view a Bayesian deep learning model as a distribution of functions. Let $p_0(f)$ be the prior distribution over the stochastic function mappings $f(\cdot; \mathbf{w})$ defined on a probability space $(\Omega, \mathcal{F}, P)$ via a

BDL model, e.g., $p_0(f)$ could be the prior distribution of a BNN, a Gaussian process or any other suitable stochastic processes. Like Bayesian inference in parameter space, the main goal is to infer the posterior over functions $p(f|\mathcal{D})$ combined with the likelihood $p(\mathbf{Y}_\mathcal{D} \mid f(\mathbf{X}_\mathcal{D}))$ evaluated at the training data $\mathcal{D} = \{\mathbf{X}_\mathcal{D}, \mathbf{Y}_\mathcal{D}\}$. However, it would be intractable for most stochastic processes. For example, as for BNNs, $p_0(f)$ is the prior distribution over functions induced by the prior distribution over random network weights $p_0(\mathbf{w})$, and there is no explicit probability form for it. Similar to parameter-space variational inference, the variational objective in function space can be denoted as $\min_{q(f;\boldsymbol{\theta_q})}\mathrm{KL}[q(f;\boldsymbol{\theta_q})\|p(f|\mathcal{D})]$, where $q(f;\boldsymbol{\theta_q})$ is the variational posterior over functions induced by the variational posterior over parameters. The functional ELBO, as the variational objective function in function space to be maximized, is

$$\mathcal{L}^{KL}_{q(f;\boldsymbol{\theta_q})} := \mathbb{E}_{q(f;\boldsymbol{\theta_q})}\left[\log p(\mathbf{Y}_\mathcal{D} \mid f(\mathbf{X}_\mathcal{D};\boldsymbol{\theta_q})\right] - \mathrm{KL}[q(f;\boldsymbol{\theta_q})\|p_0(f)], \qquad (2)$$

where one can effectively incorporate prior information about the task via the $p_0(f)$ in the KL term during the optimization. For estimation of the above functional ELBO, the following three issues need to be carefully considered:

- The first one concerns the validity of the definition of the objective function. In order to guarantee the existence of the Radon-Nikodym derivative in the KL divergence between two distributions over functions, it is necessary to satisfy that $q(f;\boldsymbol{\theta_q})$ is absolutely continuous with respect to $p_0(f)$. Specifically, for $q(f;\boldsymbol{\theta_q})$ and $p_0(f)$ generated from the same function mapping with different parameter distributions, such as the same neural network structure for BNNs, it should satisfy $\mathrm{KL}[q(\mathbf{w};\boldsymbol{\theta_q})\|p_0(\mathbf{w})] \leq \infty$ to guarantee that $\mathrm{KL}[q(f;\boldsymbol{\theta_q})\|p_0(f)] \leq \infty$ according to the strong data processing inequality(Polyanskiy & Wu, 2017) as $\mathrm{KL}[q(f;\boldsymbol{\theta_q})\|p_0(f)] \leq \mathrm{KL}[q(\mathbf{w};\boldsymbol{\theta_q})\|p_0(\mathbf{w})]$.

- The second issue concerns the implicit probability density functions for $q(f;\boldsymbol{\theta_q})$ and $p_0(f)$. Note that since $\Omega$ is infinite-dimensional function space, $p_0(f)$ and $q(f;\boldsymbol{\theta_q})$ are not actually the probability density functions but the probability measures over $\Omega$. Even for the marginal multivariate random vector $p_0(f(\mathbf{X}))$ and $q(f(\mathbf{X};\boldsymbol{\theta_q}))$ at finite input points $\mathbf{X}$, the explicit probability density functions are intractable for some stochastic processes, e.g., BNNs and other non-linear models.

- The third problem is the effective and efficient estimation of the KL divergence between two stochastic processes. To solve the intractable infinite-dimensional KL divergence between distributions over functions, Sun et al. (2019) proved that

$$\mathrm{KL}[q(f;\boldsymbol{\theta_q})\|p_0(f)] = \sup_{n\in\mathbb{N},\mathbf{X}\in\mathcal{X}^n} \mathrm{KL}[q(f(\mathbf{X};\boldsymbol{\theta_q})\|p_0(f(\mathbf{X}))], \qquad (3)$$

where $\mathcal{X}^n = \cup\{\mathbf{X} \in \mathcal{X}^n|\mathcal{X}^n \in \mathbb{R}^n\}$. In other words, the functional KL divergence is equivalent to the supremum of all KL divergence over marginal finite measurement points. Unfortunately, there is no analytical way to obtain such supremum in practical optimization.

## 3 OUR METHOD

The main obstacle in existing function-space variational inference is the definition and estimation issues regarding the KL divergence between distributions over functions. In this section, we propose a novel Wasserstein distance-based variational objective to avoid the limitations of KL divergence and improve approximation inference in function space. We firstly propose a two-step variational method via a functional prior and a bridging distribution to approximate the posterior indirectly. In the first step, we distill a functional prior by fitting a bridging distribution over functions. In the second step, we form a new ELBO by matching the variational posterior and the bridging distribution in parameter space using the 2-Wasserstein distance as a surrogate for the KL divergence. Then we further propose an integrated variational objective to jointly optimize the bridging distribution and the variational posterior.

### 3.1 FUNCTIONAL PRIOR INDUCED VARIATIONAL INFERENCE

Suppose a random function mapping $f(\cdot;\mathbf{w}) : \mathcal{X} \to \mathcal{Y}$ parametrized by random $\mathbf{w} \in \mathbb{R}^k$ for a machine learning task. The main variational objective is to obtain the approximate posterior $q(f;\boldsymbol{\theta_q})$

induced by the variational posterior over parameters $q(\mathbf{w}; \boldsymbol{\theta_q})$. Let $g(\cdot; \mathbf{w_b})$ be a latent random function with parameters $\mathbf{w_b} \in \mathbb{R}^k$ and $p(g; \boldsymbol{\theta_b})$ denote a bridging distribution over functions induced by the distribution over model parameters $p(\mathbf{w_b}; \boldsymbol{\theta_b})$. Assume that $p(g; \boldsymbol{\theta_b})$ and $q(f; \boldsymbol{\theta_q})$ are generated from the same function structure with different parametric distributions (e.g., same BNNs structure with different distributions over weights). The specific sampling process for $p(g; \boldsymbol{\theta_b})$ is as follows:

$$g(\mathbf{x}) \sim p(g|\mathbf{x}; \mathbf{w_b}), \mathbf{w_b} \sim p(\mathbf{w_b}; \boldsymbol{\theta_b}), \tag{4}$$

and the sampling process for $f$ with variational posterior $q(f; \boldsymbol{\theta_q})$ is

$$f(\mathbf{x}) \sim q(f|\mathbf{x}; \mathbf{w}), \mathbf{w} \sim q(\mathbf{w}; \boldsymbol{\theta_q}), \tag{5}$$

where $\mathbf{x} \in \mathcal{X} \subset \mathbb{R}^d$ are the input points of $f$ and $g$.

**Distilling a functional prior using a bridging distribution over functions**. Considering that GPs are well-developed priors in function space that are known to be interpretable and are able to incorporate prior knowledge about the prediction task in hand, we can assign a GP prior denoted by $p_0(f) \sim \mathcal{GP}(\mathbf{m}, \mathbf{K})$ for random $f$. Due to the intractable KL divergence between the GP prior and the non-GP variational posterior in the functional ELBO, we firstly distill the GP prior to the bridging distribution over functions by minimizing the 1-Wasserstein distance between $p(g; \boldsymbol{\theta_b})$ and $p_0(f)$ (Tran et al., 2022) with the dual form as follows:

$$W_1(p(g; \boldsymbol{\theta_b}), p_0(f)) = \sup_{\|\phi\| \leq 1} \mathbb{E}_{\mathbf{x} \sim g} \phi(\mathbf{x}) - \mathbb{E}_{\mathbf{y} \sim p_0} \phi(\mathbf{y}). \tag{6}$$

Specifically, we solve the above 1-Wasserstein distance on finite randomly sampled measurement points $\mathbf{X}_\mathcal{M} \stackrel{\text{det}}{=} [\mathbf{x}_1, \ldots, \mathbf{x}_M]^T$ as $W_1(p(g(\mathbf{X}_\mathcal{M}; \boldsymbol{\theta_b})), p_0(f(\mathbf{X}_\mathcal{M}))$ in practice due to the infinite-dimensional nature of random functions. The 1-Wasserstein distance between distributions over functions is now reduced to that over multivariate random variables. The specific form is as follows:

$$W_1(p(g(\mathbf{X}_\mathcal{M}; \boldsymbol{\theta_b})), p_0(f(\mathbf{X}_\mathcal{M})) = \sup \mathbb{E}_{\mathbf{X}_\mathcal{M}} \left[\mathbb{E}_p \phi(g(\mathbf{X}_\mathcal{M})) - \mathbb{E}_{p_0} \phi(f(\mathbf{X}_\mathcal{M}))\right], \tag{7}$$

where $\phi$ is a 1-Lipschitz continuous functions. $g(\mathbf{X}_\mathcal{M})$ and $f(\mathbf{X}_\mathcal{M})$ are corresponding function values evaluated at $\mathbf{X}_\mathcal{M}$, respectively. Not that there is another interesting work (Liu et al., 2023) uses the bi-Lipschitz condition to improve the uncertainty quality for single networks. It can be seen that this approximated computation procedure is based entirely on sampling, so it can still be performed smoothly even without the closed form of $p(g; \boldsymbol{\theta_b})$. At the same time, the functional prior-induced bridging distribution over parameters is denoted as $p(\mathbf{w_b}; \boldsymbol{\theta_b^*})$, where $\boldsymbol{\theta_b^*} = \arg\min_{\boldsymbol{\theta_b}} W_1(p(g; \boldsymbol{\theta_b}), p_0(f))$. It is worth noting that this distillation procedure could be applied to any prior distributions over functions as long as their random function samples are available.

**Matching the variational posterior and the bridging distribution in parameter space**. The main idea of this inference method is to force distribution over functions $q(f; \boldsymbol{\theta_q})$ and $p(g; \boldsymbol{\theta_b})$ to share the same function structures. That is, once the optimal functional prior-induced bridging distribution over parameters $p(\mathbf{w_b}; \boldsymbol{\theta_b^*})$ is obtained from the above distillation procedure by fitting $p_0(f)$ to $p(g; \boldsymbol{\theta_b})$, $\boldsymbol{\theta_b^*}$ is frozen and $p(\mathbf{w_b}; \boldsymbol{\theta_b^*})$ is used in the regularization term of ELBO to the variational $q(\mathbf{w}; \boldsymbol{\theta_q})$ as

$$\mathcal{L}_{q(\mathbf{w}; \boldsymbol{\theta_q})} := \mathbb{E}_{q(\mathbf{w}; \boldsymbol{\theta_q})} \left[\log p(\mathcal{D} \mid \mathbf{w}; \boldsymbol{\theta_q})\right] - \lambda W_2(q(\mathbf{w}; \boldsymbol{\theta_q}), p(\mathbf{w_b}; \boldsymbol{\theta_b^*})), \tag{8}$$

where $W_2(q(\mathbf{w}; \boldsymbol{\theta_q}), p(\mathbf{w_b}; \boldsymbol{\theta_b^*}))$ is the 2-Wasserstein distance between the approximate posterior $q(\mathbf{w}; \boldsymbol{\theta_q})$ and the bridging distribution over parameters $p(\mathbf{w_b}; \boldsymbol{\theta_b^*})$, $\lambda$ is a hyperparameter. Suppose $q(\mathbf{w}; \boldsymbol{\theta_q})$ and $p(\mathbf{w_b}; \boldsymbol{\theta_b})$ are two Gaussian distributions, then $W_2(q(\mathbf{w}; \boldsymbol{\theta_q}), p(\mathbf{w_b}; \boldsymbol{\theta_b^*}))$ has an analytical solution as

$$W_2(q(\mathbf{w}; \boldsymbol{\theta_q}), p(\mathbf{w_b}; \boldsymbol{\theta_b^*})) = \|\mu_q - \mu_b^*\|_2^2 + \text{trace}\left(\boldsymbol{\Sigma}_q + \boldsymbol{\Sigma}_b^* - 2\left(\boldsymbol{\Sigma}_q^{1/2} \boldsymbol{\Sigma}_b^* \boldsymbol{\Sigma}_q^{1/2}\right)^{1/2}\right), \tag{9}$$

where $\boldsymbol{\theta_q} := \{\mu_q, \boldsymbol{\Sigma}_q\}, \boldsymbol{\theta_b^*} := \{\mu_b^*, \boldsymbol{\Sigma}_b^*\}$ are respective mean and covariance matrices. We call this improved variational inference approach based on the functional prior the Functional Prior-induced Variational Inference (FPi-VI). A pseudocode of FPi-VI is presented in Algorithm 1 in Appendix B.

### 3.2 FUNCTIONAL WASSERSTEIN BRIDGE INFERENCE

Due to the isotropy of the 1-Wasserstein distance, there will be an infinite number of candidate $p(\mathbf{w_b}; \boldsymbol{\theta_b^*})$ with exactly same distance to a given funtional prior, and FPi-VI just randomly picks one from all candidates in the first distillation step. Such randomness brings large fluctuations to the following inference performance (see Appendix E.1). Therefore, we further treat parameters of bridging distributions $\boldsymbol{\theta_b}$ as parameters that need to be optimized together with variational posterior parameters to obtain more robust solution. Since there is no analytical solution to estimate the functional distance (e.g., KL divergence and Wasserstein distance) directly between the functional prior and variational posterior over functions for all but extremely simple distributions such as GPs, our key ingredient is to build a *Wasserstein bridge* to decompose such functional distance into a parameter-space 2-Wasserstein distance and a function-space 1-Wasserstein distance at the same time in optimization,

$$W_B(q(f; \boldsymbol{\theta_q}), p_0(f)) = \lambda_1 W_2(q(\mathbf{w}; \boldsymbol{\theta_q}), p(\mathbf{w_b}; \boldsymbol{\theta_b})) + \lambda_2 W_1\left(p(g; \boldsymbol{\theta_b}), p_0(f)\right), \quad (10)$$

where $p(\mathbf{w_b}; \boldsymbol{\theta_b})$ is the bridging distributions over parameters, $p(g; \boldsymbol{\theta_b})$ is the bridging distributions over functions, $\boldsymbol{\theta_q}$ and $\boldsymbol{\theta_b}$ are the respective stochastic parameters of the approximate variational posterior and bridging distribution which would be optimized jointly, and $\lambda_1$ and $\lambda_2$ are two hyperparameters.

Based on the Wasserstein bridge, we propose a variational objective in function space called Functional Wasserstein Bridge Inference (FWBI) and derive a practical algorithm to obtain the optimal $\{\boldsymbol{\theta_q^*}, \boldsymbol{\theta_b^*}\}$ as:

$$\{\boldsymbol{\theta_q^*}, \boldsymbol{\theta_b^*}\} = \underset{\boldsymbol{\theta_q}, \boldsymbol{\theta_b}}{\arg\min} - \mathbb{E}\left[\log p(\mathbf{Y}_\mathcal{D} \mid f(\mathbf{X}_\mathcal{D}; \boldsymbol{\theta_q}))\right] + W_B(q(f; \boldsymbol{\theta_q}), p_0(f))$$

$$= \underset{\boldsymbol{\theta_q}, \boldsymbol{\theta_b}}{\arg\min} - \frac{1}{M} \sum_{j=1}^{M} \left[\log p(\mathbf{Y}_\mathcal{B} \mid f(\mathbf{X}_\mathcal{B}; \mu_q, \boldsymbol{\Sigma}_q))\right] + \lambda_1 W_2(q(\mathbf{w}; \mu_q, \boldsymbol{\Sigma}_q), p(\mathbf{w_b}; \mu_b, \boldsymbol{\Sigma}_b)))$$

$$+ \lambda_2 W_1\left(p(g(\mathbf{X}_\mathcal{M}; \mu_b, \boldsymbol{\Sigma}_b))), p_0(f(\mathbf{X}_\mathcal{M}))\right). \quad (11)$$

where $\{\mathbf{X}_\mathcal{B}, \mathbf{Y}_\mathcal{B}\}$ is the mini-batch of the training data $\{\mathbf{X}_\mathcal{D}, \mathbf{Y}_\mathcal{D}\}$ applied to the likelihood term, $\mathbf{w}$ and $\mathbf{w_b}$ can be reparameterized as $\mathbf{w} = \mu_q + \boldsymbol{\Sigma}_q \odot \epsilon$, $\mathbf{w_b} = \mu_b + \boldsymbol{\Sigma}_b \odot \epsilon$ respectively with random noisy $\epsilon$ under the Gaussian assumption, and $\mathbf{X}_\mathcal{M}$ denotes the finite measurement set from the input space for the estimation of 1-Wasserstein distance in function space. Different from most functional variational inference approaches that perform variational optimization directly on the approximate posterior over functions, the proposed FWBI treats the bridging distribution over functions induced by distributions over parameters as an intermediate variable to link the variational posterior and functional prior. Specifically, FWBI distills a functional prior by the bridging distribution over functions $p(g; \boldsymbol{\theta_b})$ induced by $p(\mathbf{w_b}; \boldsymbol{\theta_b})$ via the 1-Wasserstein distance and matches variational posterior $q(\mathbf{w}; \boldsymbol{\theta_q})$ and $p(\mathbf{w_b}; \boldsymbol{\theta_b})$ by minimizing the 2-Wasserstein distance simultaneously. The pseudocode for FWBI is shown in Algorithm 2 in Appendix B.

The main advantages of FWBI are as follows: i) in general parameter-space mean-field variational inference, it is common to use an i.i.d Gaussian prior assumption for distributions over parameters, while FWBI can assign a more interpretable functional prior and incorporate meaningful information about the task into inference process; ii) FWBI utilizes the well-defined Wasserstein distance-based Wasserstein bridge to regularize parameters of variational posterior with a functional prior, which can circumvent the limitation of functional KL divergence used in most function-space approximate inference methods. Once the optimal $\boldsymbol{\theta_q^*}$ is obtained, the posterior predictive distribution is obtained by the following integration process and can be estimated through Monte Carlo sampling:

$$q(\mathbf{y}^*|\mathbf{x}^*) = \int p(\mathbf{y}^*|f(\mathbf{x}^*; \boldsymbol{\theta_q^*}))q(f(\mathbf{x}^*; \boldsymbol{\theta_q^*}))df(\mathbf{x}^*; \boldsymbol{\theta_q^*})$$

$$\approx \frac{1}{S} \sum_{j=1}^{S} p(\mathbf{y}^*|f(\mathbf{x}^*; \mathbf{w}^{(j)})), \mathbf{w}^{(j)} \sim \mathcal{N}(\mu_q^*, \boldsymbol{\Sigma}_q^*), \boldsymbol{\theta_q^*} := \{\mu_q^*, \boldsymbol{\Sigma}_q^*\}. \quad (12)$$

Moreover, we proved that the variational objective of FWBI is a lower bound of the model evidence, which indicates it is a well-defined objective for Bayesian inference. Based on the law of cosines

for the KL divergence and Talagrand inequality of probability measures (Appendix A), we derive the following Proposition 1 (see Appendix C for more details and proof):

**Proposition 1** *For the ELBO $\mathcal{L}^W$ derived from FWBI based on the Wasserstein bridge and the functional ELBO $\mathcal{L}^{kl}$ defined in Equation 2 based on the KL divergence, we have*

$$\log p(\mathcal{D}) \geq \mathcal{L}^{KL} \geq \mathcal{L}^W$$

*where* $\log p(\mathcal{D})$ *is the log model evidence.*

## 4 RELATED WORKS

Based on the variational inference methods in parameter spaces, there is an increasing number of works focusing on the function-space variational approaches for various BDL such as BNNs, and their applications on a range of machine learning tasks where predictive uncertainty quantification is crucial (Benjamin et al., 2019; Titsias et al., 2019; Pan et al., 2020; Rudner et al., 2022b).

**Variational inference in parameter spaces**. Parameter-space variational methods are widely used for approximating posterior over weights in BNNs. It is Hinton & Van Camp (1993) who first used variational inference in BNNs. Barber & Bishop (1998) replaced a fully factorized Gaussian assumption for variational posterior with a full rank Gaussian to model correlations between weights. Then Graves (2011) proposed a sub-sampling technique to approximate the expected log-likelihood by Monte Carlo integration. To improve this work, Blundell et al. (2015) developed an algorithm for variational inference called Bayes by Backprop (BBB) based on the reparameterization trick, which could yield an unbiased gradient estimator of ELBO w.r.t model parameters.

**Variational inference in function spaces**. Due to the limitations of parameter-space variational inference such as the intractability of specifying meaningful priors, Sun et al. (2019) proposed a kind of functional ELBO to match a GP prior and the variational posterior over functions for BNNs via a spectral Stein gradient estimator designed for implicit distributions (Shi et al., 2018). However, the KL divergence between stochastic processes involved in the functional ELBO may be ill-defined for a wide class of distributions and further leads to an invalid variational objective (Burt et al., 2020). At the same time, Wang et al. (2019) proposed a particle optimization variational inference method in function spaces for posterior approximation in BNNs. Rudner et al. (2020; 2022a) pointed out that the supremum of marginal KL divergence over finite measurement sets cannot be solved analytically for the estimation of functional KL divergence. They proposed to approximate the distributions over functions as Gaussian via the linearization of their mean parameters and derived a tractable and well-defined variational objective since the functional prior and variational posterior are two BNNs that share the same network structures. Ma & Hernández-Lobato (2021) randomize the number of finite measurement points to derive an alternative grid-functional KL divergence, which can avoid some limitations of KL divergence between stochastic processes. However, all these methods are based on KL divergence. Considering the potential weaknesses of KL divergence, there are some recent works trying to use Wasserstein distance to replace KL divergence. Tran et al. (2020) proposed to match a BNN prior to a GP prior by minimizing the 1-Wasserstein distance to obtain more interpretable functional priors in BNNs, but then they use stochastic gradient Hamiltonian Monte Carlo (SGHMC) rather than variational inference to approximate the posterior. In contrast, our work only uses the 1-Wasserstein distance to distil the functional prior and develops a functional variational objective for posterior approximation based on the proposed Wasserstein bridge. Wild et al. (2022) built a functional variational objective where the functional prior and posterior are both Gaussian measures and the dissimilarity measure was chosen to be the 2-Wasserstein distance. However, the critical GP components make it less applicable to general non-GP scenarios. In contrast, our FWBI is not restricted to any distributional assumptions: the variational posterior and prior over functions can be any reasonable stochastic process for the specific tasks.

## 5 EXPERIMENTAL EVALUATION

In this section, we evaluate the predictive performance and uncertainty quantification of FWBI on several tasks including 1-D extrapolation toy examples, multivariate regression on UCI datasets, contextual bandits, and image classification tasks. We compare FWBI to several well-established parameter-space and function-space variational inference approaches.

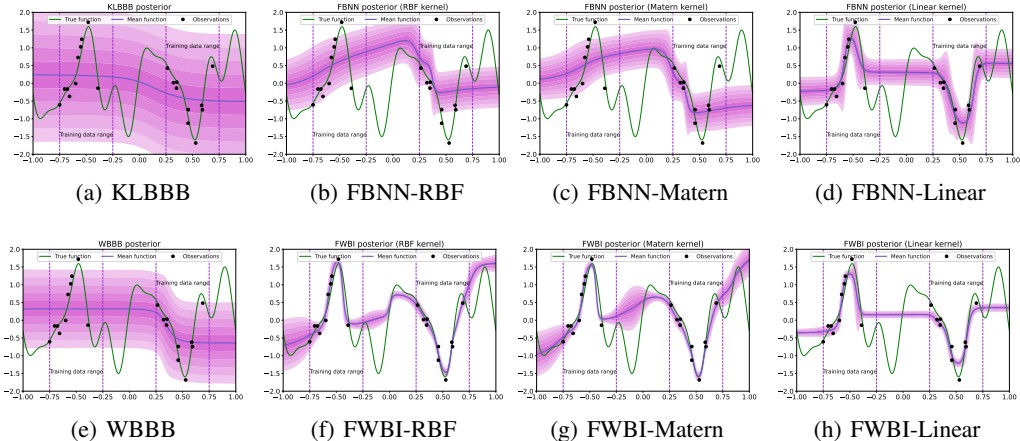

Figure 1: Learning polynomial curves. The green line is the ground true function and the blue lines correspond to mean approximate posterior predictions. Black dots denote 20 training points; shadow areas represent the predictive standard deviations. The leftmost column shows two parameter-space methods, and the other three columns are the results of functional approaches based on GP priors with three different kernels. For more details, see Appendix D.

## 5.1 Extrapolation Illustrative Examples

**Learning polynomial curves** Consider an 1-D oscillation curve from the polynomial function: $y = \sin(3\pi x) + 0.3\cos(9\pi x) + 0.5\sin(7\pi x) + \epsilon$ with noise $\epsilon \sim \mathcal{N}(0, 0.5^2)$. There are 20 randomly sampled observation points, half of which are sampled from the interval $[-0.75, -0.25]$, and the other half are from $[0.25, 0.5]$. For parameter-space variational inference comparison, we choose BBB (Blundell et al., 2015) using KL divergence for distributions over parameters, denoted by KLBBB, and a 2-Wasserstein distance alternative version called WBBB. For functional methods, we compare with the benchmark functional BNNs (FBNN) proposed by Sun et al. (2019). For FWBI and FBNN, we use the same GP prior with three different kinds of kernels: RBF kernel, Matern kernel, and Linear kernel (not suitable for modeling polynomial oscillatory curves). Results are shown in Figure 1, the leftmost column shows that the two parametric inference methods KLBBB and WBBB fail to fit the target function, while the two function-space approaches exhibit better predictive performance. For FWBI and FBNN, we first pre-train the GP prior to obtaining a more informative functional prior. Figures 1(f) and 1(g) show that FWBI is able to recover the key polynomial characteristic of the curve in observation range and provide strong uncertainty in the unseen region of input space with appropriate RBF kernel and Matern kernel. On the other hand, the mismatched Linear kernel in Figure 1(h) expresses a certain trend of error linearity, which indicates FWBI can effectively utilize functional prior information in the inference process. In contrast, FBNN under-fit the curve severely in both observations and non-observations with RBF kernel and Matern kernel, while results from the inappropriate Linear kernel are a little better. FBNN is less responsive to different kernel information and performs poorly in uncertainty estimation. See Appendix E.2 for more baseline results. Appendix E.3 shows detailed comparisons of posteriors of GPs and FWBI. The calibration curves for all methods are shown in Appendix E.4. For the analysis of the impact of functional properties of priors (smoothness and noise) on FWBI, see Appendix E.6.

**Interpolation with non-GP priors** One of the main advantages of FWBI is that it is not constrained by specific prior and variational posterior distribution families in function spaces, whether explicit or implicit, GP or non-GP. In this experiment, we consider using a BNN as the functional prior to fit a 1-D toy example: $y(x) = \sin(x) + 0.1x + \epsilon, \epsilon \sim \mathcal{N}(0, 0.5)$. 30 observations are randomly sampled from $[-7.5, -5] \cup [-2.5, 2.5] \cup [5, 7.5]$. For comparison, we also obtained the results for GP priors. As shown in Figure 2, it is obvious that FWBI has a stronger capacity to recover the key characteristics of the true function than the other inference methods with all four different priors. The BNN prior shows a competitive performance with the GP priors, and even converges faster (see Appendix E.5 for details).

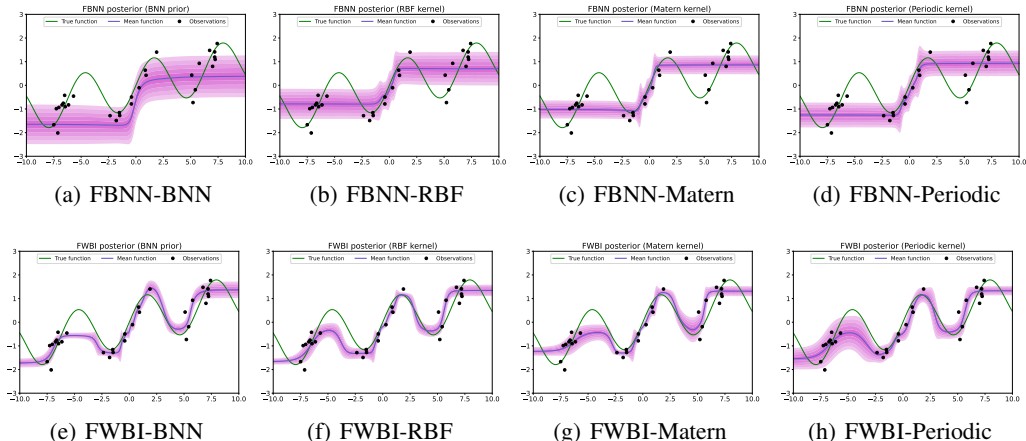

|               |               |               |               |
|:-------------:|:-------------:|:-------------:|:-------------:|
| (a) FBNN-BNN  | (b) FBNN-RBF  | (c) FBNN-Matern | (d) FBNN-Periodic |
| (e) FWBI-BNN  | (f) FWBI-RBF  | (g) FWBI-Matern | (h) FWBI-Periodic |

Figure 2: Learning with different functional priors. For the BNN prior, we use a two-hidden-layer structure. For the GP prior, we consider the RBF kernel, Matern kernel and Periodic kernel .

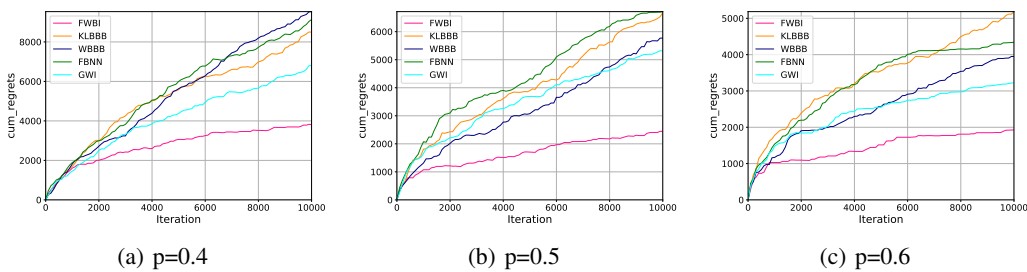

|        |        |        |
|:------:|:------:|:------:|
| (a) p=0.4 | (b) p=0.5 | (c) p=0.6 |

Figure 3: Comparisons of cumulative regrets for FWBI, KLBBB, WBBB, FBNN, GWI on the Mushroom contextual bandit task. Lower represents better performance.

Table 1: The table shows the results of average RMSE for multivariate regression on UCI datasets. We split each dataset randomly into 90% training data and 10% test data, and this process is repeated 10 times to ensure validity. We perform the paired-sample t-test for the results from FWBI and the results from other methods and get $p < .001$. See Appendix E.7 for results of the test negative log-likelihood (NLL).

| Dataset  | FWBI                  | GWI               | FBNN              | WBBB              | KLBBB             |
|----------|-----------------------|-------------------|-------------------|-------------------|-------------------|
| Yacht    | **1.249±0.090**       | $2.198 \pm 0.083$ | $1.523 \pm 0.075$ | $2.328 \pm 0.091$ | $2.131 \pm 0.085$ |
| Boston   | **1.439±0.087**       | $1.742 \pm 0.046$ | $1.683 \pm 0.122$ | $2.306 \pm 0.102$ | $1.919 \pm 0.074$ |
| Concrete | **1.072±0.068**       | $1.297 \pm 0.053$ | $1.274 \pm 0.049$ | $2.131 \pm 0.068$ | $1.784 \pm 0.063$ |
| Wine     | **1.209±0.054**       | $1.680 \pm 0.064$ | $1.528 \pm 0.053$ | $2.253 \pm 0.071$ | $1.857 \pm 0.069$ |
| Kin8nm   | **1.119±0.019**       | $1.188 \pm 0.015$ | $1.447 \pm 0.069$ | $2.134 \pm 0.029$ | $1.787 \pm 0.027$ |
| Protein  | **1.158±0.008**       | $1.333 \pm 0.007$ | $1.503 \pm 0.025$ | $2.188 \pm 0.012$ | $1.795 \pm 0.010$ |

## 5.2 MULTIVARIATE REGRESSION ON UCI DATASETS

In this experiment, we evaluate our method for multivariate regression tasks on benchmark UCI datasets to demonstrate the predictive performance of FWBI. Table 1 shows the average results of root mean square error (RMSE). All three functional inference methods consistently provide better results than parameter-space approaches, which could reflect the advantages of function-space variational inference. Furthermore, our FWBI significantly outperforms all other functional models and shows very efficient running performance (see Appendix E.8).

Table 2: Image classification and OOD detection performance.

| Model | MNIST | | FMNIST | | CIFAR10 | |
|---|---|---|---|---|---|---|
| | Accuracy | OOD-AUC | Accuracy | OOD-AUC | Accuracy | OOD-AUC |
| FWBI | $\mathbf{96.36 \pm 0.00}$ | $\mathbf{0.949 \pm 0.03}$ | $\mathbf{85.85 \pm 0.00}$ | $\mathbf{0.838 \pm 0.01}$ | $\mathbf{46.62 \pm 0.01}$ | $0.616 \pm 0.03$ |
| GWI | $95.40 \pm 0.00$ | $0.858 \pm 0.05$ | $85.43 \pm 0.00$ | $0.394 \pm 0.04$ | $44.78 \pm 0.01$ | $\mathbf{0.635 \pm 0.02}$ |
| FBNN | $96.09 \pm 0.00$ | $0.801 \pm 0.07$ | $85.64 \pm 0.00$ | $0.814 \pm 0.02$ | $46.29 \pm 0.01$ | $0.612 \pm 0.03$ |
| WBBB | $96.16 \pm 0.00$ | $0.869 \pm 0.03$ | $85.57 \pm 0.00$ | $0.819 \pm 0.01$ | $45.76 \pm 0.01$ | $0.606 \pm 0.03$ |
| KLBBB | $96.26 \pm 0.00$ | $0.868 \pm 0.03$ | $85.71 \pm 0.00$ | $0.829 \pm 0.02$ | $46.20 \pm 0.00$ | $0.606 \pm 0.02$ |

## 5.3 CONTEXTUAL BANDITS

Reliable uncertainty estimation is crucial for downstream tasks such as contextual bandit problems, where the agent gradually learns the model by observing a context repeatedly and choosing the action with the lowest regrets in dynamic environments. In these scenarios, it is important to balance the exploration and exploitation during the optimization. In this section, we evaluate the ability of FWBI to guide exploration on the UCI Mushroom dataset, which includes 8124 instances, and each mushroom has 22 features and is identified as edible or poisonous. The agent can observe these mushroom features as the context and choose either to eat or reject a mushroom to maximize the reward. We consider three different reward patterns: for the action of eating a mushroom if the mushroom is edible, the agent will receive a reward of 5. Conversely, if the mushroom is poisonous, the agent will receive a reward of -35 with probabilities 0.4, 0.5, and 0.6 respectively for three different patterns, otherwise a reward of 5. On the other hand, if the agent decides to take the action of rejecting a mushroom, it will receive a reward of 0.

Suppose an oracle will always choose to eat an edible mushroom (and receive a reward of 5) and not to eat the poisonous mushroom. We take the cumulative regrets with respect to the reward achieved by the oracle to measure the exploration-exploitation ability of an agent. We concatenate the mushroom context and the action chosen by the agent as model input and the corresponding received reward is the model output. We follow the hyperparameter settings by Blundell et al. (2015). The cumulative regrets of all 5 parameter-space and function-space variational inference methods for 3 reward patterns are shown in Figure 3. FWBI performs significantly better than other inference methods in all three reward modes, which indicates that FWBI is able to provide reliable uncertainty estimation in such decision-making scenarios.

## 5.4 CLASSIFICATION AND OOD DETECTION

We evaluate the scalability of FWBI via image classification tasks with high-dimensional inputs. We assess the in-distribution predictive performance and out-of-distribution (OOD) detection ability on MNIST, FashionMNIST(Xiao et al., 2017) and CIFAR-10(Krizhevsky et al., 2009). For all inference methods, we use fully-connected BNNs as variational posteriors and a GP prior for functional inference methods. We report the test accuracy for predictive performance and the area under the curve (AUC) of OOD detection pairs FashionMNIST/MNIST, MNIST/FashionMNIST and CIFAR10/SVNH based on predictive entropies in Table 2. Our FWBI consistently outperforms all parameter-space and function-space baselines for classification accuracy and performs competitively in OOD detection (Appendix E.9).

## 6 CONCLUSION

In this paper, we proposed a new function-space variational inference method termed Functional Wasserstein Bridge Inference (FWBI). It optimizes a Wasserstein bridge-based functional variational objective as the surrogate to the possible problematic KL divergence between stochastic processes involved in most existing functional variational inference. We proved the functional ELBO derived from FWBI is a lower bound of the model evidence. Empirically, we demonstrated that FWBI can leverage various GP or non-GP functional priors to yield high predictive performance and principled uncertainty quantification.

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

## A  FURTHER BACKGROUND

**Pathologies for parameter-space priors**  As in Figure 4, we show the function samples generated from three BNNs with Gaussian prior $\mathcal{N}(0, 1)$ over network weights. It is obvious that as the depth increases, the function samples tend to be more horizontal, which can lead to a problematic posterior inference.

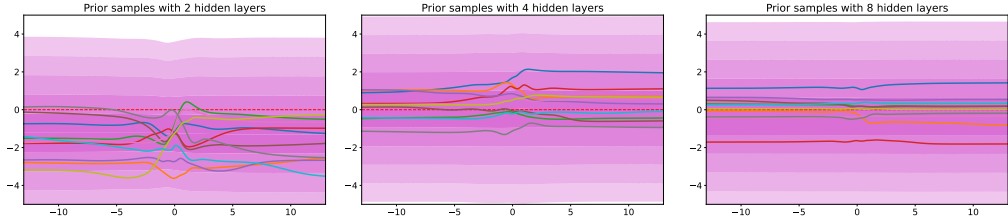

Figure 4: Function samples from three fully-connected BNNs with different network architectures: there are 2, 4, and 8 hidden layers respectively, and each layer with 50 units. The prior distribution for weights is $\mathcal{N}(0, 1)$ and the activation is tanh.

**Wasserstein distance**  The *Wasserstein distance* (Kantorovich, 1960; Villani, 2003) is a rigorously defined distance metric on probability measures satisfying non-negativity, symmetry and triangular inequality (Panaretos & Zemel, 2019) that was originally proposed for the optimal transport problem and has become popular in the machine learning community in recent years (Arjovsky et al., 2017). Suppose $(\mathcal{P}, \|\cdot\|)$ is a Polish space, the p-Wasserstein distance between probability measures $\mu$, $\nu \in (\mathcal{P}, \|\cdot\|)$ is defined as

$$W_p(\mu, \nu) = \left(\inf_{\gamma \in \Gamma(\mu, \nu)} \int_{\mathcal{P} \times \mathcal{P}} \|x - y\|^p \, \mathrm{d}\gamma(x, y)\right)^{1/p}, \tag{13}$$

where $\Gamma(\mu, \nu)$ is the set of joint measures or coupling $\gamma$ with marginals $\mu$ and $\nu$ on $\mathcal{P} \times \mathcal{P}$.

**The law of cosines for the KL divergence**  For two probability measures $p$ and $q \in (\mathcal{P}, \|\cdot\|)$, the law of cosines for the KL divergence between $p$ and $q$ is defined as (Belavkin, 2013):

$$\begin{aligned}
\mathrm{KL}[p\|q] &= \mathrm{KL}[p\|r] + \mathrm{KL}[r\|q] - \int \log \frac{dq(x)}{dr(x)}[dp(x) - dr(x)] \\
&= \mathrm{KL}[p\|r] - \mathrm{KL}[q\|r] - \int \log \frac{dq(x)}{dr(x)}[dp(x) - dq(x)].
\end{aligned} \tag{14}$$

where $r$ is the reference measure. Consider the 1-Wasserstein distance between $p$ and $q$ as

$$W_1(p, q) := \sup\{\mathbb{E}_p\{f\} - \mathbb{E}_q\{g\} : f(x) - g(y) \le c(x, y)\}. \tag{15}$$

Suppose real function $f(x)$ and $g(x)$ satisfying additional constraints:

$$\beta f(x) = \nabla \mathrm{KL}[p\|r] = \log \frac{dp(x)}{dr(x)}, \quad \beta \ge 0 \tag{16}$$

$$\alpha g(x) = \nabla \mathrm{KL}[q\|r] = \log \frac{dq(x)}{dr(x)}, \quad \alpha \ge 0 \tag{17}$$

Thus, $\beta f$ and $\alpha g$ are the gradients of divergence $\mathrm{KL}[p\|r]$, $\mathrm{KL}[q\|r]$ respectively, and this means that probability measures $p, q$ have the following exponential representations:

$$dp(x) = e^{\beta f(x) - \kappa[\beta f]} dr(x) \tag{18}$$

$$dq(x) = e^{\alpha g(x) - \kappa[\alpha g]} dr(x) \tag{19}$$

where $\kappa[(\cdot)] = \log \int e^{(\cdot)} dr(x)$ is the normalizing constant.

$$\frac{d}{d\beta}\kappa[\beta f] = \mathbb{E}_p\{f\}, \quad \mathrm{KL}[p\|r] = \beta \mathbb{E}_p\{f\} - \kappa[\beta f] \tag{20}$$

$$\frac{d}{d\alpha}\kappa[\alpha g] = \mathbb{E}_q\{g\}, \quad \mathrm{KL}[q\|r] = \alpha\mathbb{E}_q\{g\} - \kappa[\alpha g] \tag{21}$$

Substituting these formulate into equation 14 we obtain

$$\mathrm{KL}[p\|q] = \beta\mathbb{E}_p\{f\} - \alpha\mathbb{E}_q\{g\} - (\kappa[\beta f] - \kappa[\alpha g]) - \alpha\int g(x)[dp(x) - dq(x)] \tag{22}$$

According to the Theoreom 2 in Belavkin (2018), assume that Lagrange multipliers $\alpha = \beta = 1$, then have

$$\mathrm{KL}[p\|q] = \mathbb{E}_p\{f\} - \mathbb{E}_q\{g\} - (\kappa[f] - \kappa[g]) - \int g(x)[dp(x) - dq(x)]$$
$$= W_1(p, q) - (\kappa[f] - \kappa[g]) - \int g(x)[dp(x) - dq(x)] \tag{23}$$

**Talagrand inequality**   As proved by Otto & Villani (2000), the probability measure $q$ satisfies a Talagrand inequality with constant $\rho$ if for all probability measure $p$, absolutely continuous w.r.t. $q$, with finite moments of order 2,

$$W_1(p, q) \leq W_2(p, q) \leq \sqrt{\frac{2\mathrm{KL}[p\|q]}{\rho}} \tag{24}$$

where the first inequality can be proved by the Cauchy-Schwarz inequality.

## B   PSEUDOCODE OF FPI-VI AND FWBI

---
**Algorithm 1** Functional Prior-induced Variational Inference (FPi-VI)

---
**Require:** Dataset $\mathcal{D} = \{\mathbf{X}_\mathcal{D}, \mathbf{Y}_\mathcal{D}\}$, minibatch $\{\mathbf{X}_\mathcal{B}, \mathbf{Y}_\mathcal{B}\} \subset \mathcal{D}$, functional prior $p_0(f)$
 1: Initialise $\mathbf{w} \sim \mathcal{N}(0, 1)$, $\mathbf{w}_b \sim \mathcal{N}(0, 1)$, reparameterize $\mathbf{w} = \mu_q + \boldsymbol{\Sigma}_q \odot \epsilon$, $\mathbf{w_b} = \mu_b + \boldsymbol{\Sigma}_b \odot \epsilon$ with $\epsilon \sim \mathcal{N}(0, 1)$, $\boldsymbol{\theta_q} := \{\mu_q, \boldsymbol{\Sigma}_q\}$, $\boldsymbol{\theta_b} := \{\mu_b, \boldsymbol{\Sigma}_b\}$
 2: **while** $\theta_b$ not converged **do**
 3:     draw measurement set $\mathbf{X}_\mathcal{M}$ randomly from input domain
 4:     draw functional prior functions $f(\mathbf{X}_\mathcal{M}) \sim p_0(f)$ at $\mathbf{X}_\mathcal{M}$
 5:     draw bridging distribution functions $g(\mathbf{X}_\mathcal{M}) \sim p(g; \boldsymbol{\theta_b})$ at $\mathbf{X}_\mathcal{M}$
 6:     calculate $W_1\left(p(g(\mathbf{X}_\mathcal{M}; \mu_b, \boldsymbol{\Sigma}_b))), p_0(f(\mathbf{X}_\mathcal{M}))\right)$ using Equation 7
 7:     $\boldsymbol{\theta_b} \leftarrow \mathrm{Optimizer}(\boldsymbol{\theta_b}, \mathrm{W}_1)$
 8: **end while**
 9: Froze $\boldsymbol{\theta_b^*} := \{\mu_b^*, \boldsymbol{\Sigma}_b^*\}$
10: **while** $\theta_q$ not converged **do**
11:     calculate $\mathcal{L} = -\frac{1}{M}\sum_{j=1}^M \left[\log p(\mathbf{Y}_\mathcal{B} \mid f(\mathbf{X}_\mathcal{B}; \mu_q, \boldsymbol{\Sigma}_q)] + \lambda W_2(q(\mathbf{w}; \boldsymbol{\theta_q}), p(\mathbf{w}_b; \boldsymbol{\theta_b^*}))\right.$ using Equation 9
12:     $\boldsymbol{\theta_q} \leftarrow \mathrm{Optimizer}(\boldsymbol{\theta_q}, \mathcal{L})$
13: **end while**

---

## C   PROOF OF THEORETICAL RESULTS

To analyze the theoretical properties of FWBI for posterior variational inference, we further derive a new corresponding functional ELBO as follows:

$$\mathcal{L}^W := \mathbb{E}\left[\log p(\mathbf{Y}_\mathcal{D} \mid f(\mathbf{X}_\mathcal{D}; \boldsymbol{\theta_q})\right] - \lambda_1 W_2(q(f; \boldsymbol{\theta_q}), p(g; \boldsymbol{\theta_b})) - \lambda_2 W_1\left(p(g; \boldsymbol{\theta_b}), p_0(f)\right), \tag{25}$$

where $q(f; \boldsymbol{\theta_q})$ is the variational distribution over functions induced by approximate posterior $q(\mathbf{w}; \boldsymbol{\theta_q})$. $W_2(q(f; \boldsymbol{\theta_q}), p(g; \boldsymbol{\theta_b}))$ is calculated by corresponding $W_2(q(\mathbf{w}; \boldsymbol{\theta_q}), p(\mathbf{w}_b; \boldsymbol{\theta_b}))$. As a variational Bayesian objective, it is worthwhile to explore whether this new ELBO based on Wasserstein bridge is still a lower bound of the log marginal likelihood. Based on the law of cosines for the KL divergence and Talagrand inequality of probability measures (Appendix A), we derive the following Proposition 1:

---

**Algorithm 2** Functional Wasserstein Bridge Inference (FWBI)

---

**Require:** Dataset $\mathcal{D} = \{\mathbf{X}_{\mathcal{D}}, \mathbf{Y}_{\mathcal{D}}\}$, minibatch $\{\mathbf{X}_{\mathcal{B}}, \mathbf{Y}_{\mathcal{B}}\} \subset \mathcal{D}$, functional prior $p_0(f)$

1: Initialise Initialise $\mathbf{w} \sim \mathcal{N}(0, 1)$, $\mathbf{w}_b \sim \mathcal{N}(0, 1)$, reparameterize $\mathbf{w} = \mu_q + \Sigma_q \odot \epsilon$, $\mathbf{w}_b = \mu_b + \Sigma_b \odot \epsilon$ with $\epsilon \sim \mathcal{N}(0, 1)$, $\boldsymbol{\theta_q} := \{\mu_q, \Sigma_q\}$, $\boldsymbol{\theta_b} := \{\mu_b, \Sigma_b\}$

2: **while** $\boldsymbol{\theta_b}, \boldsymbol{\theta_q}$ not converged **do**

3:     draw measurement set $\mathbf{X}_{\mathcal{M}}$ randomly from input domain

4:     draw functional prior functions $f(\mathbf{X}_{\mathcal{M}}) \sim p_0(f)$ at $\mathbf{X}_{\mathcal{M}}$

5:     draw bridging distribution functions $g(\mathbf{X}_{\mathcal{M}}) \sim p(g; \boldsymbol{\theta_b})$ at $\mathbf{X}_{\mathcal{M}}$

6:     $\mathcal{L} = -\frac{1}{M} \sum_{j=1}^{M} [\log p(\mathbf{Y}_{\mathcal{B}} \mid f(\mathbf{X}_{\mathcal{B}}; \mu_q, \Sigma_q)] + \lambda_1 W_2(q(\mathbf{w}; \mu_q, \Sigma_q), p(\mathbf{w}_b; \mu_b, \Sigma_b))) + \lambda_2 W_1(p(g(\mathbf{X}_{\mathcal{M}}; \mu_b, \Sigma_b))), p_0(f(\mathbf{X}_{\mathcal{M}})))$

7:     $\boldsymbol{\theta_b}, \boldsymbol{\theta_q} \leftarrow \text{Optimizer}(\boldsymbol{\theta_b}, \boldsymbol{\theta_q}, \mathcal{L})$

8: **end while**

---

**Proposition 1** *For the ELBO $\mathcal{L}^W$ derived from FWBI based on the Wasserstein bridge and the functional ELBO $\mathcal{L}^{kl}$ defined in Equation 2 based on the KL divergence, we have*

$$\log p(\mathcal{D}) \geq \mathcal{L}^{KL} \geq \mathcal{L}^W$$

*where $\log p(\mathcal{D})$ is the log model evidence.*

*Proof.* Since the 1-Wasserstein distance is not greater than the 2-Wasserstein distance between two probability measures based on the Cauchy-Schwarz inequality, we first have:

$$\mathcal{L}^W := \mathbb{E}[\log p(\mathbf{Y}_{\mathcal{D}} \mid f(\mathbf{X}_{\mathcal{D}}; \boldsymbol{\theta_q})] - \lambda_1 W_2(q(f; \boldsymbol{\theta_q}), p(g; \boldsymbol{\theta_b})) - \lambda_2 W_1(p(g; \boldsymbol{\theta_b}), p_0(f))$$

$$\leq \mathbb{E}[\log p(\mathbf{Y}_{\mathcal{D}} \mid f(\mathbf{X}_{\mathcal{D}}; \boldsymbol{\theta_q})] - \lambda_1(\text{KL}[q(f; \boldsymbol{\theta_q}) \| p(g; \boldsymbol{\theta_b})] + \int g_1(f)[dq(f; \boldsymbol{\theta_q}) - dp(g; \boldsymbol{\theta_b})]) -$$

$$\lambda_2(\text{KL}[p(g; \boldsymbol{\theta_b}) \| p_0(f)] + \int g_2(f)[dp(g; \boldsymbol{\theta_b}) - dp_0(f)])$$

$$= \mathbb{E}[\log p(\mathbf{Y}_{\mathcal{D}} \mid f(\mathbf{X}_{\mathcal{D}}; \boldsymbol{\theta_q})] - (\lambda_1(\text{KL}[q(f; \boldsymbol{\theta_q}) \| p(g; \boldsymbol{\theta_b})]) + \lambda_2(\text{KL}[p(g; \boldsymbol{\theta_b}) \| p_0(f)])) -$$

$$(\lambda_1 \int g_1(f)[dq(f; \boldsymbol{\theta_q}) - dp(g; \boldsymbol{\theta_b})] + \lambda_2 \int g_2(f)[dp(g; \boldsymbol{\theta_b}) - dp_0(f)])$$

$$= \mathbb{E}[\log p(\mathbf{Y}_{\mathcal{D}} \mid f(\mathbf{X}_{\mathcal{D}}; \boldsymbol{\theta_q})] - \text{KL}[q(f; \boldsymbol{\theta_q}) \| p_0(f)] - (\int \log \frac{dp_0(f)}{dp(g; \boldsymbol{\theta_b})}[dq(f; \boldsymbol{\theta_q}) - dp(g; \boldsymbol{\theta_b})] +$$

$$\int \log \frac{dp(g; \boldsymbol{\theta_b})}{dr(f)}[dq(f; \boldsymbol{\theta_q}) - dp(g; \boldsymbol{\theta_b})] + \int \log \frac{dp_0(f)}{dr(f)}[dp(g; \boldsymbol{\theta_b}) - dp_0(f)])$$

$$= \mathbb{E}[\log p(\mathbf{Y}_{\mathcal{D}} \mid f(\mathbf{X}_{\mathcal{D}}; \boldsymbol{\theta_q})] - \text{KL}[q(f; \boldsymbol{\theta_q}) \| p_0(f)] - (\int \log \frac{dp_0(f)}{dr(f)}[dq(f; \boldsymbol{\theta_q}) - dp(g; \boldsymbol{\theta_b})] +$$

$$\int \log \frac{dp_0(f)}{dr(f)}[dp(g; \boldsymbol{\theta_b}) - dp_0(f)])$$

$$= \mathbb{E}[\log p(\mathbf{Y}_{\mathcal{D}} \mid f(\mathbf{X}_{\mathcal{D}}; \boldsymbol{\theta_q})] - \text{KL}[q(f; \boldsymbol{\theta_q}) \| p_0(f)] - \int \log \frac{dp_0(f)}{dr(f)}[dq(f; \boldsymbol{\theta_q}) - dp_0(f)]$$

$$= \mathcal{L}^{KL} - \int \log \frac{dp_0(f)}{dr(f)}[dq(f; \boldsymbol{\theta_q}) - dp_0(f)]$$

$$= \mathcal{L}^{KL} - \int \log \frac{dp_0(f)}{-dq(f; \boldsymbol{\theta_q})}[dq(f; \boldsymbol{\theta_q}) - dp_0(f)]$$

$$= \mathcal{L}^{KL} - (\text{KL}[q(f; \boldsymbol{\theta_q}) \| p_0(f)] + \text{KL}[p_0(f) \| q(f; \boldsymbol{\theta_q})])$$

$$\leq \mathcal{L}^{KL}$$

$$\leq \log p(\mathcal{D})$$

$$\tag{26}$$

where we assume that $\lambda_1 = \lambda_2 = 1$ and the reference measure $dr(f) = -dq(f; \boldsymbol{\theta_q})$ in the law of cosines for the KL divergence.

Proposition 1 shows that $\mathcal{L}^W$ is a valid variational objective function since it is a lower bound of the model evidence.

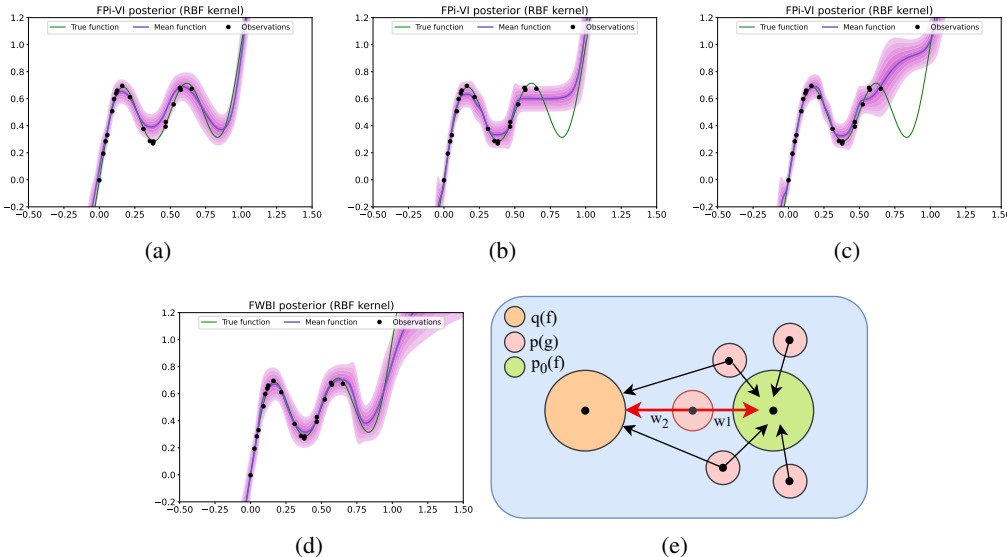

Figure 5: Explanation for potential sub-optimal solution in FPi-VI.

# D  EXPERIMENTAL SETTING

**Polynomial curve extrapolation** In this experiment, we use $2 \times 100$ fully connected tanh BNNs as variational posteriors for all models. The functional GP priors are pre-trained on the test dataset (1000 data points in $[-1, 1]$) for 100 epochs. We also use 40 inducing points for the sampling of marginal measurement points in FWBI, FBNN and GWI from $[-1, 1]$. All methods are trained for 10000 epochs.

**Interpolation with non-GP priors** In this example, the functional BNN prior has two hidden layers, each with 100 units. The BNN prior and GP priors are both pre-trained on the test data (1000 data points in $[-10, 10]$) for 100 epochs. The marginal measurement set for FWBI and FBNN is randomly sampled from all observations together with 30 inducing points randomly sampled from $[-10, 10]$. All inference methods are trained for 10000 epochs.

**Multivariate regression on UCI datasets** We choose BNNs posteriors with two hidden layers (input-10-10-output). The GP prior uses RBF kernel and is pre-trained on the test dataset for 100 epochs. The number of iterations for all models is 2000.

**Contextual bandits** The variational posteriors are fully connected tanh BNNs with two hidden layers (input-100-100-output) and the GP prior is pre-trained on 1000 randomly sampled points from training data. ALL models are trained using the last 4096 input-output tuples in the training buffer with a batch size of 64 and training frequency 64 for each iteration. All inference methods are trained for 10000 epochs.

**Classification and OOD detection** For all models in this experiment, the variational posteriors are fully connected BNNs with 2 hidden layers, each with 800 units. The functional prior is Dirichlet-based GP designed for classification tasks Milios et al. (2018) and is pre-trained on test dataset for 500 epochs. ALL inference methods are trained for 600 epochs and the batchsize is 125.

# E  FURTHER RESULTS

## E.1  DEMONSTRATION FOR POTENTIAL SUB-OPTIMAL FPI-VI

Due to the isotropy of the 1-Wasserstein distance, there will be an infinite number of candidate $p(\mathbf{w_b}; \boldsymbol{\theta_b^*})$ with exactly the same distance to a given functional prior, and FPi-VI just randomly picks one from all candidates in the first distillation step as shown in Figure 5(e). Such randomness brings

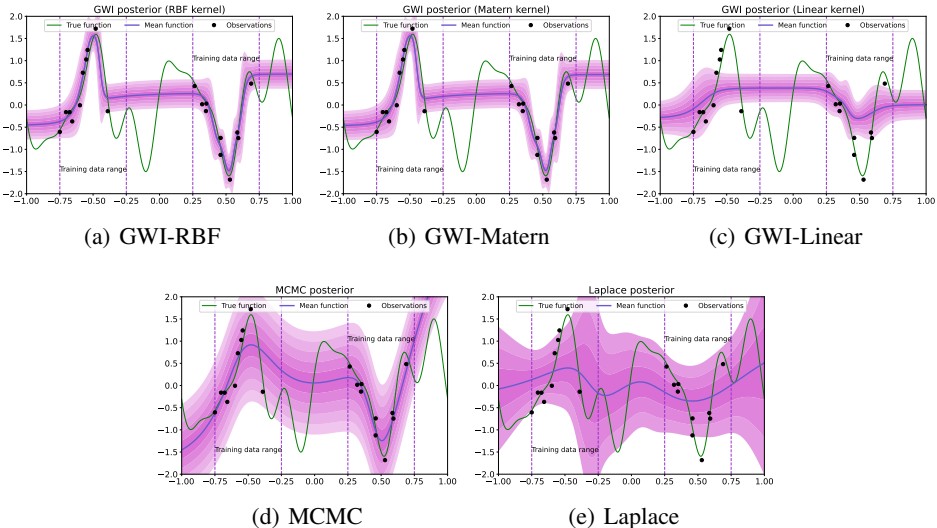

Figure 6: More baseline results for polynomial extrapolation example.

large fluctuations to the following inference performance. To show such a problem, we first train FWBI on a 1-D toy example (results are shown in Figure 5(d)), the output distance of the learned bridging distribution to the GP prior is evaluated and denoted as $d = 0.0335$. Then, we use the 1-Wasserstein distance as the loss to (randomly) find another three $p(\mathbf{w_b}; \boldsymbol{\theta_b^*})$ with the same distance $d$ to the GP prior[1], and we run the second step of FPi-VI using the three $p(\mathbf{w_b}; \boldsymbol{\theta_b^*})$ respectively. The results are plotted in Figures 5(a), 5(b) and 5(c). We can observe that 1) the performances of all models are significantly different even though their bridging distributions all have the same distance to the prior, which demonstrates that the first step of FPi-VI is with large fluctuations may harm the final posterior inference; 2) FWBI is much better than the other three, which shows that FWBI could automatically learn a reasonably good bridging distribution from the infinite number of candidates.

### E.2 MORE BASELINE RESULTS FOR TOY EXAMPLE

Figure 6 shows more baseline results for the toy example. Figure 6(a), 6(b) and 6(c) are the approximate posteriors of GWI using three different kernels corresponding to which used in Figure 1. And Figure 6(d) is the mean of samples from MCMC posterior using Langevin dynamics and 6(e) is the posterior from Laplace approximationDaxberger et al. (2021). Compared to these baseline results, FWBI still shows stronger ability to recover the main trend of the target function and competitive uncertainty estimation in the unseen region.

### E.3 DETAILED COMPARISONS WITH GP POSTERIORS FOR TOY EXAMPLE

In this section, we give a detailed comparison between the FWBI posterior and the GP posterior. We first pre-trained GP priors with the Matern kernel and the RBF kernel on 20 training data points, and the results for the corresponding GP posteriors and FWBI posteriors are given in Figure 7. In Figure 7(a), the GP posterior shows excessive uncertainty in the well-fitted training region, which is barely distinguishable from the uncertainty in the unseen region. In contrast, our model is able to achieve more reasonable uncertainty estimates. As shown in Figure 7(c) for FWBI posterior, in the well-fitted intervals $[-0.75, -0.25]$ and $[0.25, 0.75]$ containing training points, the uncertainty is significantly smaller than that in the three regions without data. For the results corresponding to the RBF kernel in Figure 7(b) and 7(d), both models show good uncertainty estimation at the same time, however, our FWBI posterior demonstrates better fitting ability, e.g., in the middle unseen region $[-0.25, 0.25]$, our model better recovers the trend of the objective function. It is worth mentioning that in the previous experiment in Figure1, in order to give a prior that is as accurate as possible, we

---

[1]In the implementation, we allow a small variation to it as $d + \epsilon$ where $\epsilon < 0.001$

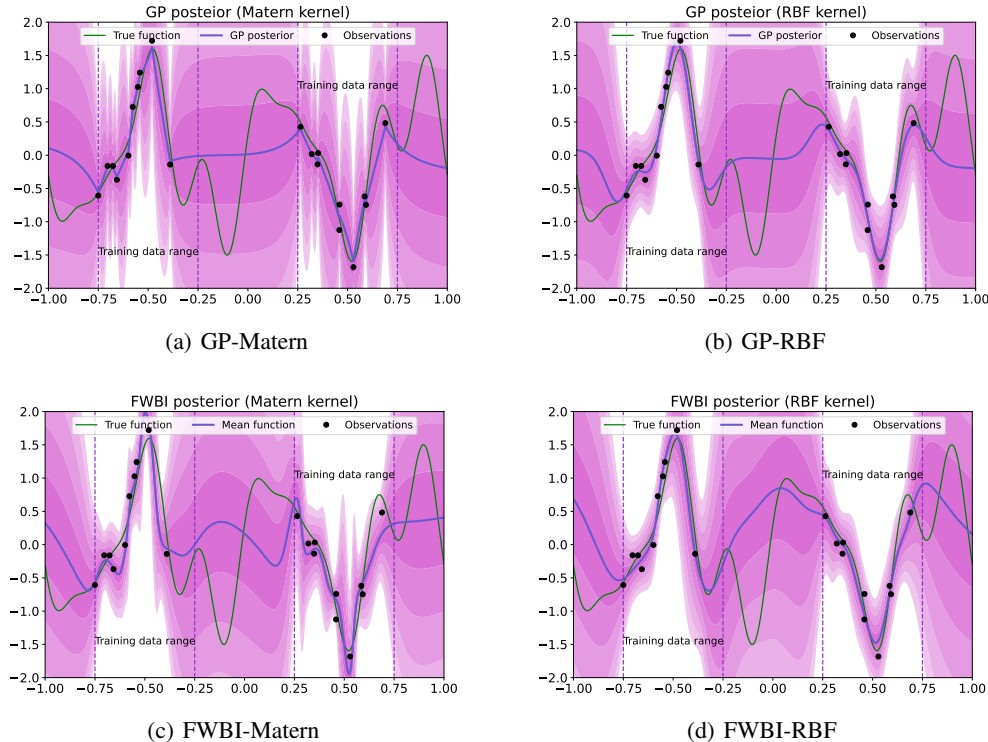

Figure 7: Comparisons of posteriors of GP and FWBI.

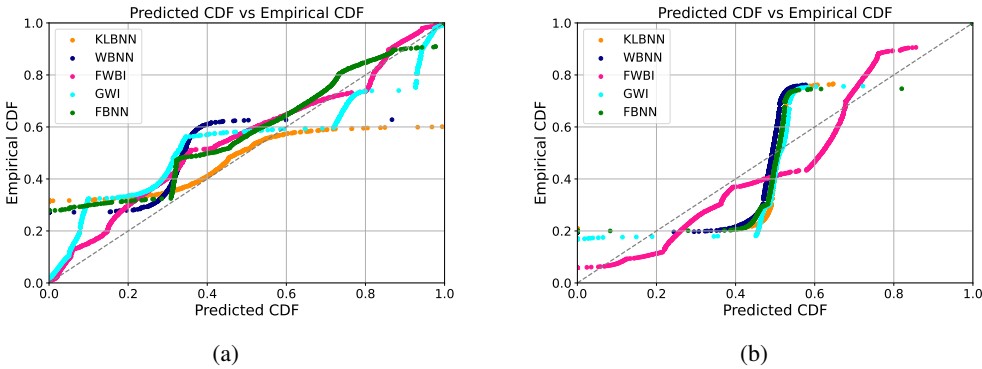

Figure 8: Calibration curves for two toy examples. The gray dashed line is the perfect calibration.

used all the test data to pre-train GP priors (for all methods), so the uncertainty estimations of the obtained corresponding posteriors are relatively small.

### E.4 CALIBRATION CURVES FOR TOY EXAMPLES

Referring to the calibration curve for regression tasks in Kuleshov et al. (2018), it can measure how well the predicted probabilities match the observed frequencies. As shown in Figure 8, we plot calibration curves of all methods for two toy examples, where the horizontal and vertical coordinates are the predicted cumulative distribution function (CDF) and the empirical CDF, respectively. The 45-degree diagonal line represents the perfect calibration, where we can see that our FWBI shows the most superior calibration.

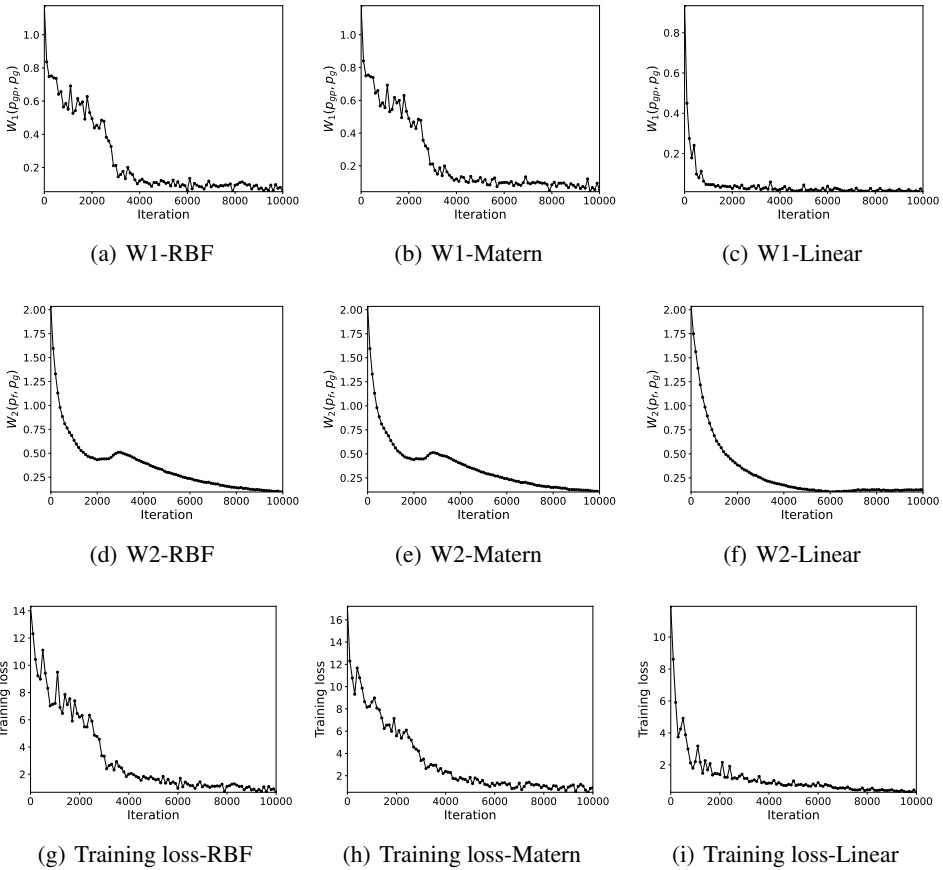

Figure 9: Convergence of Wasserstein bridge in the training of FWBI for polynomial extrapolation.

### E.5 CONVERGENCES OF WASSERSTEIN BRIDGE IN FWBI

The convergence processes of 1-Wasserstein distance and 2-Wasserstein of FWBI in two toy examples are shown in Figure 9 and Figure 10. As shown in Figure 10(a), 10(b) and 10(c), $W_1$ in BNN prior convergences faster than that in GP prior for this toy example, therefore, as shown in Figure 11(a), 11(b) and 11(c), FWBI with BNN prior performs better than that with GP priors at the 8000 epoch.

### E.6 IMPACT OF FUNCTIONAL PROPERTIES OF PRIOR ON FWBI

In this section, we analyze the impact of functional properties (smoothness and noise) of prior on FWBI posterior in Appendix E.6. Consider a 1-D periodic function: $y = 2 * \sin(4x) + \epsilon$ with noise $\epsilon \sim \mathcal{N}(0, 0.01)$, and randomly sample 20 training points from this function within $[-2, -0.5] \cup [0.5, 2]$. Firstly, we fit a GP with Matern kernel as the prior using these training data. Since the Matern kernel has a parameter $\nu$ used to control the smoothness of the functions from GP, we use $\nu$ to simulate different prior smoothness. The results are shown in Figure 12, where Figure 12(a) and 12(b) are the results from the unsmoothed ($\nu = 0.5$) and smoothed ($\nu = 2.5$) GP priors, respectively. The corresponding FWBI posteriors are shown in Figures 12(c) and 12(d), where we can see the smoothness of the prior has very limited effects on the resulting posteriors.

Then, we investigate the impact of prior noise on the posteriors by adding different GP noises to the pre-trained GP prior (with periodic kernel). Specifically, we consider two situations: GP noises with fixed 0 mean and varying variances, and GP noises with fixed variance and varying means, respectively. The results are shown in Figures 13 and 14, where the left column shows several GP priors with different injected noises, and the right column shows the corresponding FWBI posteriors.

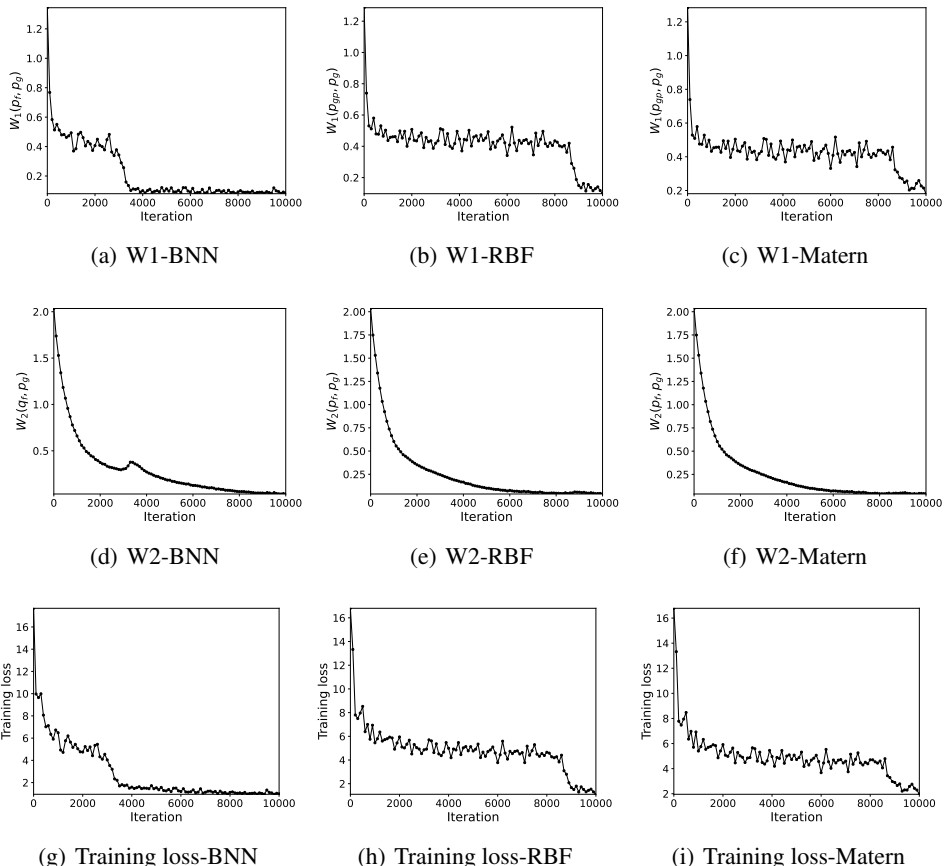

(a) W1-BNN      (b) W1-RBF      (c) W1-Matern

(d) W2-BNN      (e) W2-RBF      (f) W2-Matern

(g) Training loss-BNN      (h) Training loss-RBF      (i) Training loss-Matern

Figure 10: Convergence of Wasserstein bridge in the training of FWBI for interpolation with non-GP prior example.

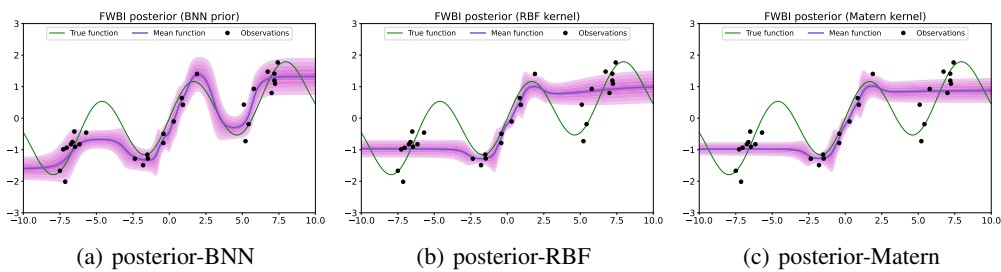

(a) posterior-BNN      (b) posterior-RBF      (c) posterior-Matern

Figure 11: The corresponding approximate posteriors of FWBI at 8000 epochs for different functional priors.

We can observe that: 1) when the mean of noises is fixed, there is no significant effect from varying variances (from 0.5 to 5) on the region with training data and only a minor effect on the right-hand side region without observed training data. The larger variances tend to destroy the prediction on the non-data regions; 2) when the variance of noises is fixed, varying means (from 0.3 to 3) also have little effect on the region with training data but destroy the prediction on the non-data regions. The larger changed means would lead to worse prediction.

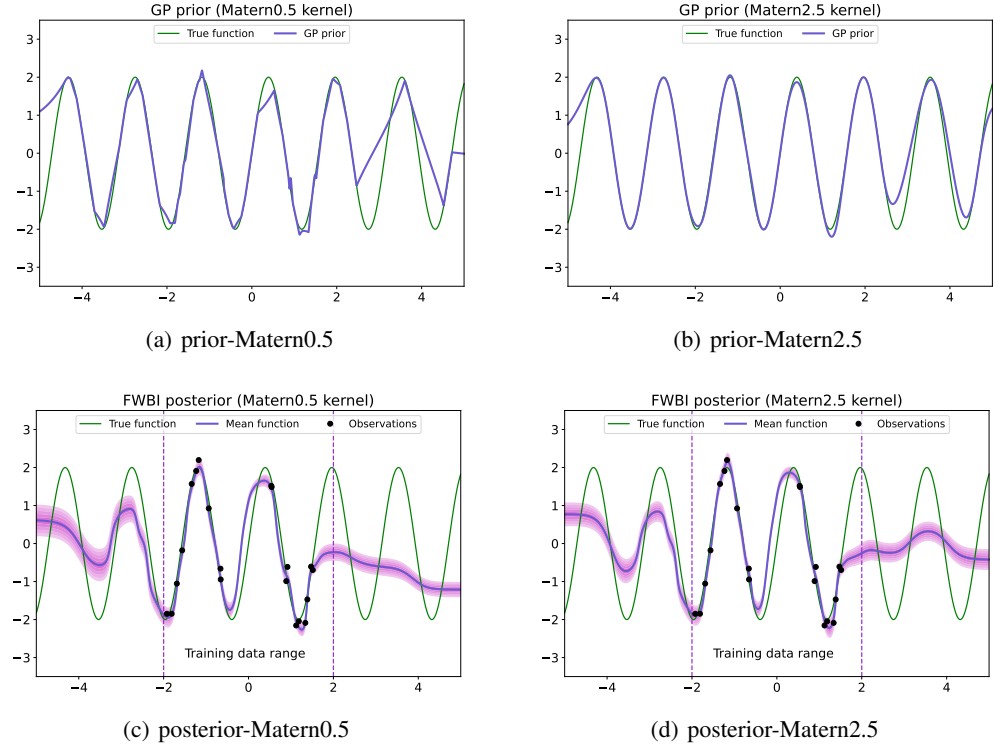

Figure 12: The effect of the prior smoothness on FWBI posterior.

Table 3: The table shows the average test NLL on several UCI regression tasks. We split each dataset randomly into 90% of training data and 10% of test data. This process is repeated 10 times to ensure validity.

| Dataset | FWBI | GWI | FBNN | WBBB | KLBBB |
|---|---|---|---|---|---|
| Yacht | **-1.249±1.214** | $0.112 \pm 0.757$ | $-0.770 \pm 0.869$ | $2.856 \pm 0.186$ | $2.512 \pm 0.161$ |
| Boston | $0.324 \pm 0.304$ | $-1.043 \pm 0.681$ | **-1.193±0.763** | $2.656 \pm 0.179$ | $2.066 \pm 0.115$ |
| Concrete | $-0.394 \pm 0.335$ | $-0.684 \pm 0.492$ | **-1.001±0.520** | $2.838 \pm 0.152$ | $2.614 \pm 0.166$ |
| Wine | **0.259±0.151** | $0.700 \pm 0.159$ | $0.524 \pm 0.137$ | $2.843 \pm 0.147$ | $2.148 \pm 0.125$ |
| Kin8nm | $-1.014 \pm 0.139$ | **-2.604±0.237** | $-2.445 \pm 0.622$ | $2.823 \pm 0.066$ | $2.614 \pm 0.071$ |
| Protein | **-2.126±0.340** | $-1.575 \pm 0.229$ | $-1.486 \pm 0.238$ | $2.744 \pm 0.026$ | $2.222 \pm 0.020$ |

### E.7 NLL Results for UCI Regressions

Table 3 shows the average test negative log likelihood (NLL) results on UCI regression tasks. FWBI still shows competitive performance compared to other weight-space and functional methods.

### E.8 Running Time Comparison for UCI Regressions

In order to compare the efficiency between FWBI and other inference approaches, we provide the running time on a small Boston dataset and a large Protein dataset in multivariate regression tasks. For Boston dataset, it has 455 training points with 13 dimensional features, while there are 41157 training points with 9 input dimensions in the larger Protein dataset. The GPU running time for all 2000 training epochs of each method is shown in Table 4.

We can see that in a small Boston dataset, the running time of FWBI is similar to parameter-space WBBB and KLBBB, and FWBI is nearly 10 times faster than FBNN. And for large Protein datasets, the running time of FBNN and GWI is 5-7× higher than FWBI, which indicates that FWBI is very

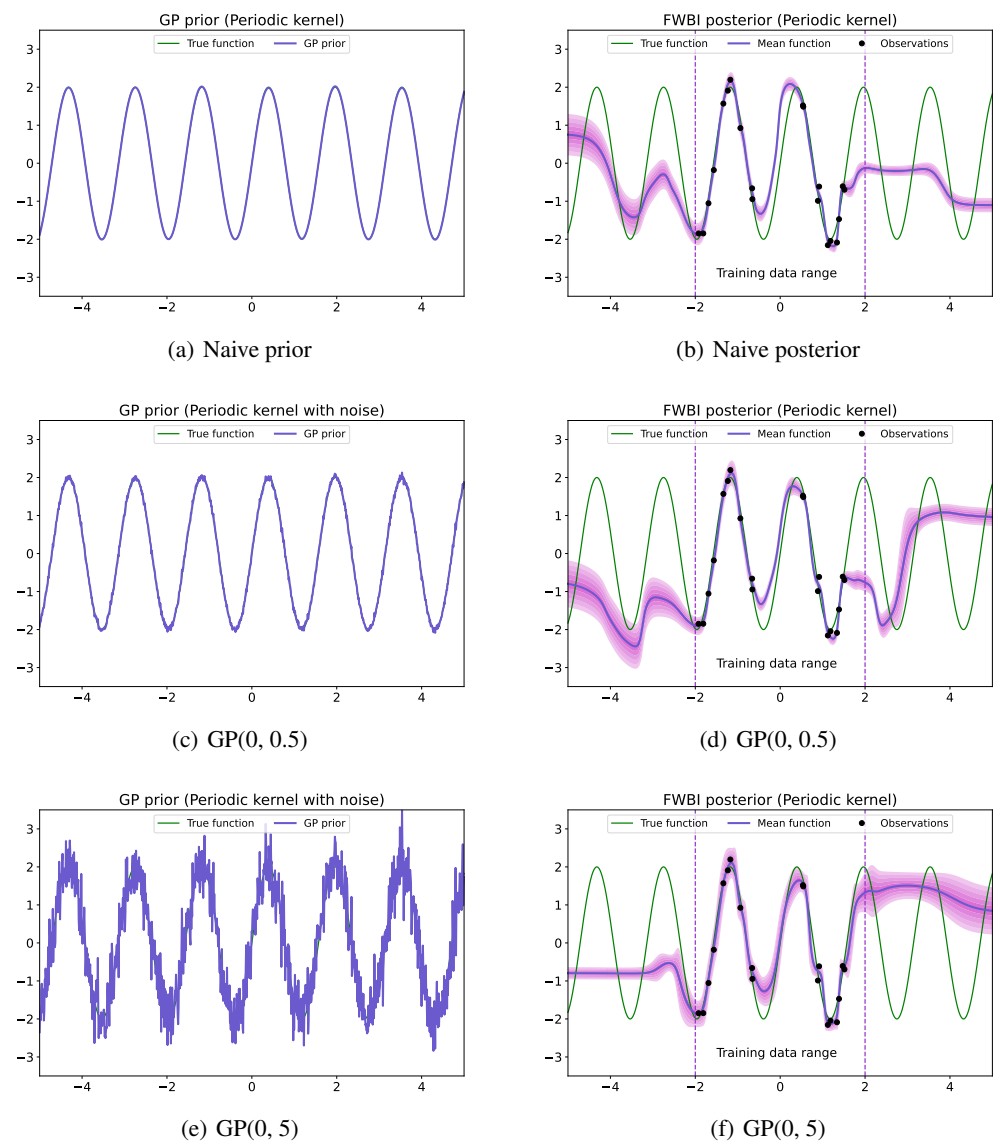

Figure 13: The effect of the prior noise with fixed mean on FWBI posterior (subtitles represent the injected noise). The top row is the naive GP prior and the corresponding FWBI posterior. The left column is the GP priors with different injected GP noises, and the right column is the corresponding FWBI posteriors.

Table 4: Running time comparison on Boston and Protein dataset.

| Run time(s) | FWBI | GWI | FBNN | WBBB | KLBBB |
|---|---|---|---|---|---|
| Boston | $15.67 \pm 0.577$ | $16.00 \pm 1.000$ | $200.67 \pm 6.351$ | $8.33 \pm 0.577$ | $8.33 \pm 0.577$ |
| Protein | $73.33 \pm 3.055$ | $472.67 \pm 0.577$ | $318.33 \pm 5.774$ | $8.00 \pm 0.000$ | $8.00 \pm 1.000$ |

efficient. Additionally, the convergence processes of training loss for all methods are shown in Figure 15, FWBI shows significant advantages in terms of both convergence speed and stability.

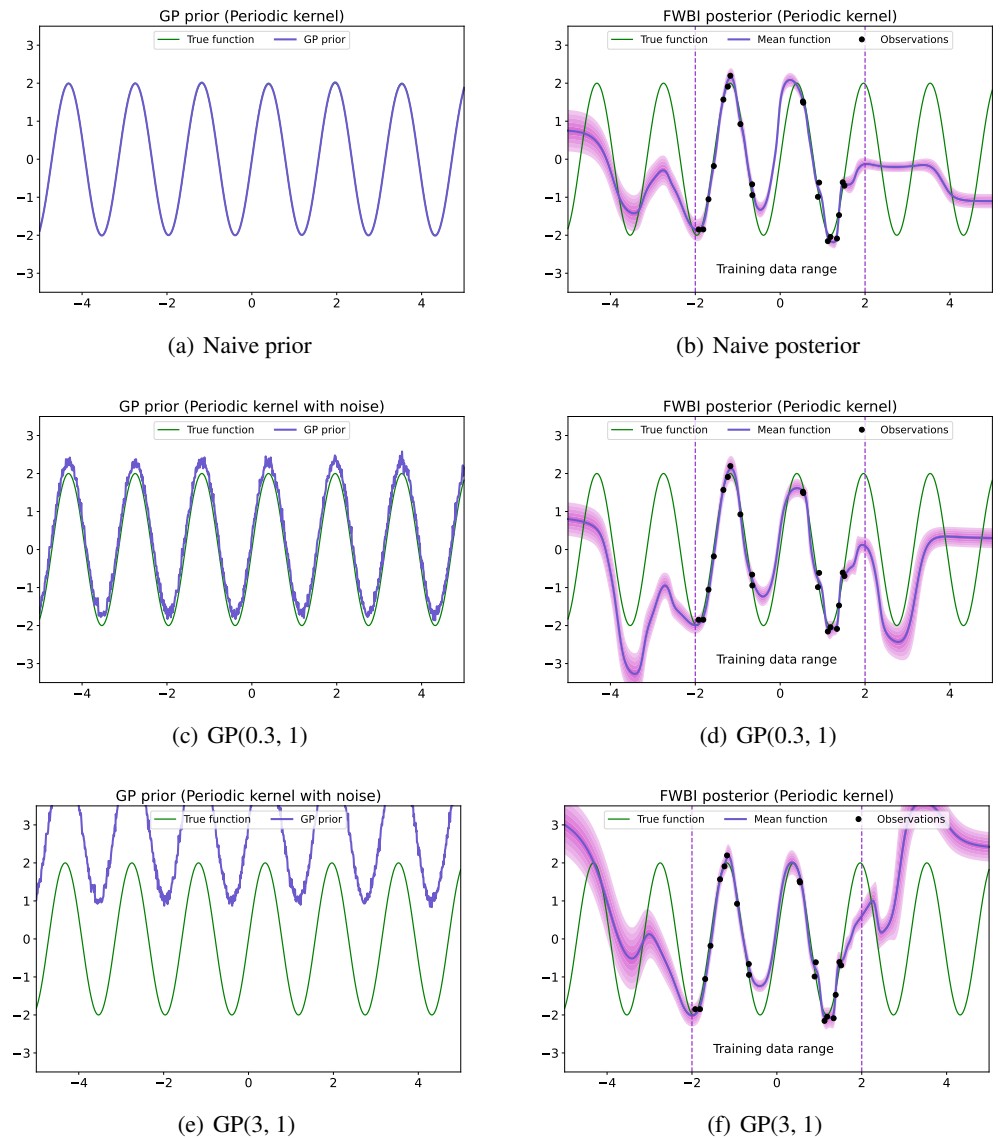

Figure 14: The effect of the prior noise with fixed variance on FWBI posterior (subtitles represent the injected noise). The top row is the naive GP prior and the corresponding FWBI posterior. The left column is the GP priors with different injected GP noises, and the right column is the corresponding FWBI posteriors.

## E.9    ROC FOR OOD DETECTION IN CLASSIFICATION TASKS

Figure 16 shows the receiver operating characteristic curve (ROC) for all methods on OOD detection in image classification tasks. The closer the curve is to the upper left corner, the stronger the OOD detection capability. Our FWBI performs competitively in all three datasets.

## E.10    WASSERSTEIN DISTANCE VS. KL DIVERGENCE

We first define a Gaussian mixture model (GMM) as our target distribution,

$$p(x) = 0.1 * \mathcal{N}\left(x; \begin{bmatrix} 0 \\ 0 \end{bmatrix}, \begin{bmatrix} 2 & 0 \\ 0 & 2 \end{bmatrix}\right) + 0.2 * \mathcal{N}\left(x; \begin{bmatrix} 20 \\ 20 \end{bmatrix}, \begin{bmatrix} 3 & 0 \\ 0 & 3 \end{bmatrix}\right) + 0.7 * \mathcal{N}\left(x; \begin{bmatrix} -10 \\ 20 \end{bmatrix}, \begin{bmatrix} 1 & 0 \\ 0 & 1.5 \end{bmatrix}\right)$$

(27)

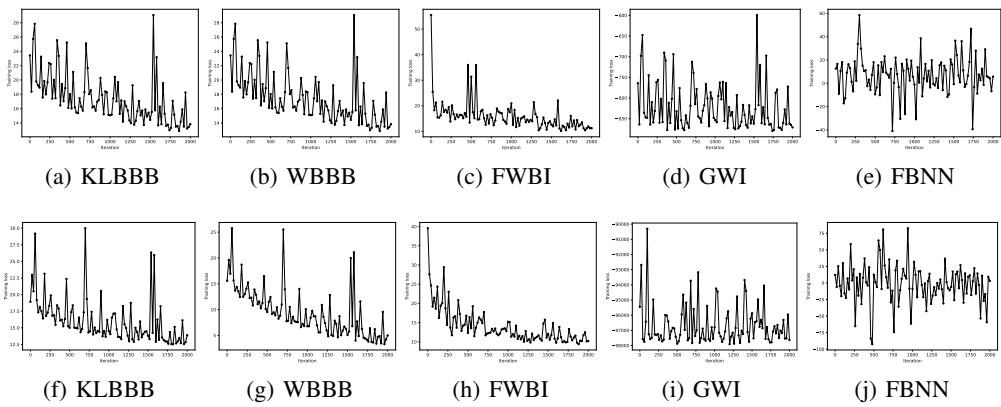

(a) KLBBB  (b) WBBB  (c) FWBI  (d) GWI  (e) FBNN

(f) KLBBB  (g) WBBB  (h) FWBI  (i) GWI  (j) FBNN

Figure 15: Convergence of training loss for multivariate regression tasks. The top row are results for the Boston dataset, and the bottom row are results for the Protein dataset.

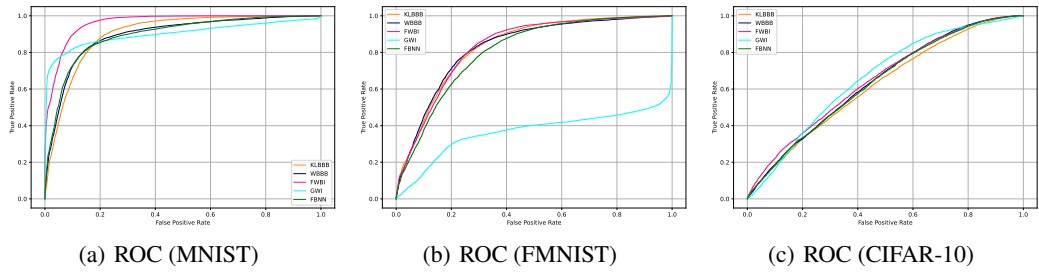

(a) ROC (MNIST)  (b) ROC (FMNIST)  (c) ROC (CIFAR-10)

Figure 16: Receiver operating characteristic curve (ROC) for out-of-distribution detection.

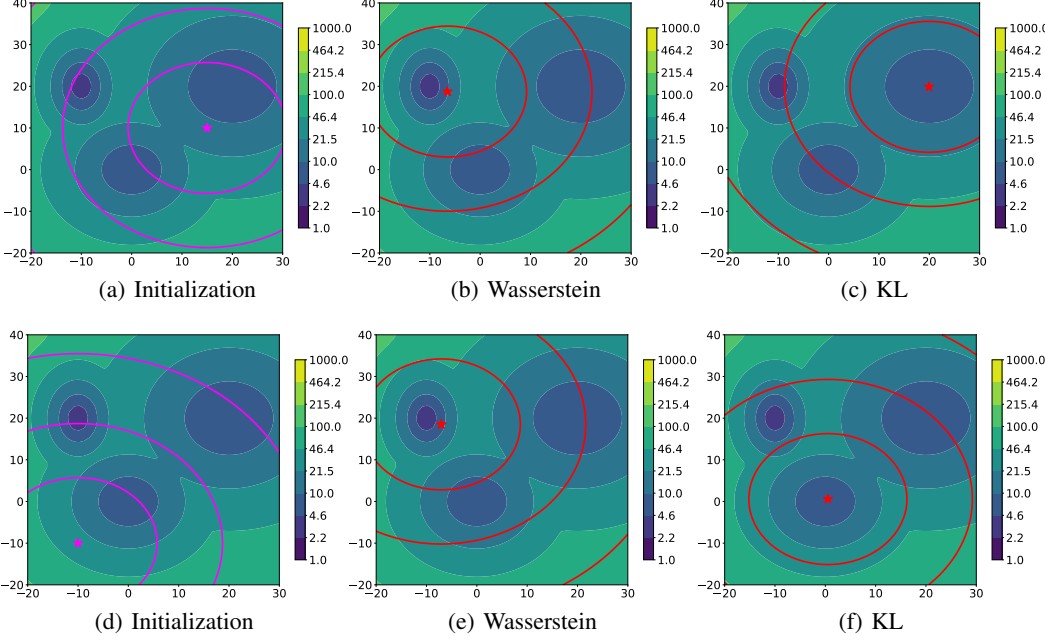

(a) Initialization  (b) Wasserstein  (c) KL

(d) Initialization  (e) Wasserstein  (f) KL

Figure 17: Approximation results from different loss. The background contour field is a Gaussian mixture with three components.

Table 5: Notation table

| Notation | Meanings |
|---|---|
| $\mathcal{D} = \{\mathbf{X}_{\mathcal{D}}, \mathbf{Y}_{\mathcal{D}}\}$ | Training dataset |
| $\mathcal{X} \subseteq \mathbb{R}^d$ | ($d$-dimensional) input space |
| $\mathcal{Y} \subseteq \mathbb{R}^c$ | ($c$-dimensional) output space |
| $\mathbf{X}$ | Finite marginal points |
| $\mathbf{X}_{\mathcal{M}}$ | Finite measurement points |
| $\mathbf{w} \in \mathbb{R}^k$ | Random model parameters for a BDL model (e.g., network weights of a BNN) |
| $\mathbf{w}_b \in \mathbb{R}^k$ | Random model parameters for a latent function |
| $f(\cdot; \mathbf{w})$ | Random function mapping defined by a BDL model (e.g., a BNN) parameterized by $\mathbf{w}$ |
| $g(\cdot; \mathbf{w}_b)$ | Random latent function parameterized by $\mathbf{w}_b$ |
| $\boldsymbol{\theta}_q = \{\mu_q, \boldsymbol{\Sigma}_q\}$ | Parameters for variational distribution |
| $\boldsymbol{\theta}_b = \{\mu_b, \boldsymbol{\Sigma}_b\}$ | Parameters for bridging distribution |
| $p_0(\mathbf{w})$ | Prior distribution over model parameters (e.g., prior over weights in a BNN) |
| $p(\mathbf{w}|\mathcal{D})$ | Posterior over model parameters (e.g., posterior over weights in a BNN) |
| $q(\mathbf{w}; \boldsymbol{\theta}_q)$ | Variational posterior over model parameters (e.g., variational posterior over weights in a BNN) |
| $p(\mathbf{w}_b; \boldsymbol{\theta}_b)$ | Bridging distribution over parameters |
| $p_0(f)$ | Prior distribution over random functions |
| $p(f|\mathcal{D})$ | Posterior over functions |
| $q(f; \boldsymbol{\theta}_q)$ | Variational posterior over functions |
| $p(g; \boldsymbol{\theta}_b)$ | Bridging distribution over functions |

where three components are included with corresponding weights. The log-likelihood contour field is plotted in Figure 17. We then use a Gaussian distribution

$$q(x) = 0.1 * \mathcal{N}\left(x; \mu, \begin{bmatrix} 5 & 0 \\ 0 & 5 \end{bmatrix}\right) \tag{28}$$

to approximate the above-defined GMM distribution, where $\mu$ is the mean parameter that needs to be optimized. Finally, we use KL divergence ($KL[q||p]$) and Wasserstein distance ($W_1[q, p]$) as the loss function to optimize $\mu$, respectively. All other hyperparameters are the same for all, like optimizer, steps, and learning rates.

The results are shown in Figure 17, where we set two different initializations (Figures 17(a) and 17(d)). We can see that

- KL divergence is sensitive to initialization. For different initializations, there are two different results (Figures 17(c) and 17(f)) from KL divergence. In contrast, the results from Wasserstein distance (Figures 17(b) and 17(e)) are the same under different initializations.

- Wasserstein could jump out of the local optimum and move close to the global optimal mode (which is the up-left corner one with the darkest colour in Figure 17).

## F    NOTATION TABLE

Table 5 is the notation table to demonstrate the notation used in this paper.

