# OpenReview forum: "Functional Wasserstein Bridge Inference for Bayesian Deep Learning"
_ICLR.cc/2024/Conference — Submitted to ICLR 2024_

### Official Review · Reviewer_kp8E · 2023-10-20

**Soundness:** 2 fair
**Presentation:** 2 fair
**Contribution:** 3 good
**Rating:** 3
**Confidence:** 4

**Summary:**

Bayesian neural networks (BNNs) are powerful for uncertainty quantification. However, the connection between the parameter space and the function space of NNs is highly nontrivial---it is thus hard to impose an interpretable prior over functions on BNNs. Functional variational BNNs (FVBNNs) can alleviate this issue. However, they are defined through the KL-divergence which leads to some pathologies. The main issue is that the KL-divergence-based variational objective is often ill-defined for FVBNNs since absolute continuity between the variational posterior and the prior is needed.

To alleviate these issues, the authors propose to "translate" functional BNNs problems into parametric ones. First, they match the (arbitrary) functional prior to the same parametric family as the variational posterior (e.g. Gaussians) by minimizing the 1-Wasserstein distance, approximated using samples. Then, they use the resulting parametric prior for standard parametric variational inference. These two steps can be collapsed into one, resulting in the Functional Wasserstein Bridge Inference (FWBI) objective.

Experimental results show that FWBI is better than previous KL-based FVBNNs.

**Strengths:**

- FWBI as a method is sound and backed by a guarantee (Prop. 1) that it results in a valid lower bound to the marginal likelihood.
- FWBI achieves better results than previous FVBNNs.

**Weaknesses:**

There are two main issues from my point of view. I'm happy to increase my score if they are sufficiently addressed.

**Baseline selection**

While the experiment setup is good (it encompasses different applications such as regression, bandits, and classification), it ignores non-variational baselines, such as the deep kernel learning (DKL, Wilson et al., 2016) and the Laplace approximation (LA, Daxberger et al., 2021). They are both function-space BNNs---for the latter, it is given by the linearized (Immer et al., 2021) and the last-layer Laplace (Riquelme et al., 2018) formulations. They have been shown to be very scalable (for the last-layer version, even applicable to transformers) and yield good uncertainty quantification performance. Moreover, have also been applied in similar experimental setups as in the present paper (e.g. bandits (see [Sec. 3.2 here](https://arxiv.org/pdf/2310.00137.pdf)) or [Bayesian optimization](https://arxiv.org/abs/2304.08309)). So I think comparing the FWBI with a Laplace baseline is a must, both in terms of performance and costs. The latter is very important since it dictates whether a method is practical or not.

Regarding Tab. 2, the classification performance presented is very bad. I suggest the authors use more standard networks for those problems. E.g. LeNet-5 for MNIST and FMNIST, and ResNet for CIFAR-10. Otherwise, Tab. 2 will only cast FWBI in a bad light.

**Unclear contributions**

The authors mention that "prior distillation"---the first step in FWBI---and "Wasserstein variational inference"---the second step of FWBI, have been done separately before. This makes FWBI seem to simply combine two prior things together. Could the authors comment on this?

The "Related Work" section is supposed to address this confusion. However, I find that that section simply describes prior works without discussing the differences between them and the authors' present work. I suggest the authors to rework that section for better clarity.

**Questions:**

First, some minor suggestions: I believe Algorithm 2 and Prop. 1 should not be in the appendix. I think they are integral parts of the paper and can be very useful for the readers to gain intuition about the proposed method.

Some questions:

1. Prop. 1 tells us that the FWBI objective is a lower bound to the standard ELBO. While it is good that FWBI defines a lower bound to the marginal likelihood, I think the result is incomplete. For example, a sufficiently small constant function will also be a valid lower bound, but obviously, it is a bad lower bound. Could you please comment on how good is the FWBI objective?

2. Still in Prop. 1: Ignoring the obvious flaws presented in Sec. 2, can you comment on why FWBI yields better performance than other (KL-based) FVBNNs even though the KL-ELBO is a tighter lower bound?

---

> ### Author Response · Authors · 2023-11-22
> **Response part one for Reviewer kp8E**
>
> We express our gratitude to Reviewer kp8E for positive and insightful feedback. In response to your comments, we prepared a revised version of the manuscript and address specific questions below.
>
> > Q1: Prop. 1 tells us that the FWBI objective is a lower bound to the standard ELBO. While it is good that FWBI defines a lower bound to the marginal likelihood, I think the result is incomplete. For example, a sufficiently small constant function will also be a valid lower bound, but obviously, it is a bad lower bound. Could you please comment on how good is the FWBI objective?
>
> $\textbf{Reply:}$ We want to firstly explain the role of ELBO here. The reason why we want to have a valid ELBO of the data likelihood is because we can achieve the marginal likelihood maximization via maximizing the valid ELBO. Yes, a sufficiently small constant function could also be a lower bound for marginal likelihood, but we cannot achieve the marginal likelihood maximization via maximizing this “sufficiently small constant function”. Maximizing a good lower bound could improve the data marginal likelihood and then improve the predictive performance, so we can use our experimental results to evaluate our FWBI objective.
>
> > Q2: Still in Prop. 1: Ignoring the obvious flaws presented in Sec. 2, can you comment on why FWBI yields better performance than other (KL-based) FVBNNs even though the KL-ELBO is a tighter lower bound?
>
> $\textbf{Reply:}$ Firstly, the tighter bower is preferred due to the larger possibility of obtaining maximized data likelihood, but it does not guarantee to obtain better performance in practice. One example is Trust Region Policy Optimization (TRPO) [1] which uses KL divergence to replace the Total Variation divergence in practice but, in fact, Total Variation divergence has a tighter bound than KL.
>
> Secondly, apart from the obvious flaws presented in Section 2, one possible reason is that KL divergence is easily caught in a local minimum, but Wasserstein distance has the better ability to jump out of local minimum. To show that, we add a Figure 17 in Appendix E.10, where we use KL and Wasserstein distances as the loss function to approximate a Gaussian mixture model (with three components) using a (single-mode) Gaussian distribution. We found that 1) KL divergence is sensitive to the initialization. For different initializations, there are two different results (Figures 17(c) and 17(f)) from KL divergence. In contrast, the results from Wasserstein distance (Figures 17(b) and 17(e)) are the same under different initializations; and 2) Wasserstein could jump out of the local optimum and move close to the global optimal mode (which is the up-left corner one with the darkest colour in Figure 17).
>
> [1], https://arxiv.org/pdf/1502.05477.pdf

---

> ### Author Response · Authors · 2023-11-22
> **Response part two for Reviewer kp8E**
>
> $\textbf{Baseline selection } $
>
> > While the experiment setup is good (it encompasses different applications such as regression, bandits, and classification), it ignores non-variational baselines, such as the deep kernel learning (DKL, Wilson et al., 2016) and the Laplace approximation (LA, Daxberger et al., 2021).
>
> $\textbf{Reply:}$ For deep kernel learning, unlike our FWBI, we think its main application is not to BNNs, but to some tasks about GPs, so we did not compare FWBI to it for the time being. For Laplace approximation (LA, Daxberger et al., 2021), we have added this baseline results on the toy example in Figure 6 in Appendix E.2.  Compared to it, our model shows stronger fitting ability and competitive uncertainty estimation.
>
> > Regarding Tab. 2, the classification performance presented is very bad. I suggest the authors use more standard networks for those problems. E.g. LeNet-5 for MNIST and FMNIST, and ResNet for CIFAR-10. Otherwise, Tab. 2 will only cast FWBI in a bad light.
>
> $\textbf{Reply:}$ Yes, we may obtain better performance using the other more sophisticated network architectures, and so will other methods. Other possible ways to further improve the performance may include batch normalization, data augmentation, adaptive learning rate, hyperparameter tuning using black-box optimizer, etc. However, the classification experiment here is used to demonstrate the effectiveness of the proposed functional inference compared with other methods. We think a fair setting for all comparative methods is enough.
>
> > Unclear contributions
>
> $\textbf{Reply:}$ We revised the ‘Related Work’ section to emphasize our innovations and contributions compared to prior works. We did use the similar idea with “prior distillation”, but we want to highlight that “prior distillation” is not for functional variational inference. The task of “prior distillation” is the prior matching. The other difference is that the 1-Wasserstein distance to measure the distance between two stochastic processes is only one terms of our algorithm, and we have another two terms: one is log-likelihood and the other is the distance between variational posterior and bridging distribution. The three terms are jointly optimized in a functional variational inference framework which is our main contribution in this study. “Wasserstein variational inference” is firstly a parameter-space method but ours is functional space method. Secondly, “Wasserstein variational inference” defines the loss function using the Wasserstein distance between variational distribution and prior, but ours is a log-likelihood term plus the Wasserstein distance between variational distribution with bridging distribution. The two forms are not equal with each other because the distance is Wasserstein rather than KL divergence.
>
> We would like to thank Reviewer kp8E, and we hope that our changes adequately address your concerns. Please let us know if you have any further questions or comments, and we are very happy to follow up!

---

> > ### Comment · Reviewer_kp8E · 2023-11-22
> > **Reply**
> >
> > Thanks for your responses! I only have one single additional comment:
> >
> > > Yes, we may obtain better performance using the other more sophisticated network architectures, and so will other methods. Other possible ways to further improve the performance may include batch normalization, data augmentation, adaptive learning rate, hyperparameter tuning using black-box optimizer, etc. However, the classification experiment here is used to demonstrate the effectiveness of the proposed functional inference compared with other methods. We think a fair setting for all comparative methods is enough.
> >
> > Your first sentence is of course correct. But the key here is how costly and how easy are those methods applied to NNs that are relevant practice. It is fine to have a somewhat expensive method, as long as the tradeoff (in terms of performance) is worth it. For this reason, I asked you to compare with Laplace in different (not toy) setups: Laplace is good and cheap for many tasks like image classification [[1]](https://arxiv.org/abs/2106.14806), LLM language modeling [[2]](https://arxiv.org/abs/2308.13111), contextual bandits [[3]](https://arxiv.org/abs/2310.00137), and Bayesian optimization [[4]](https://arxiv.org/abs/2304.08309)---it can thus serve as a strong baseline against your method, which in turn makes your argument much stronger.
> >
> > In any case, I will assign my final score after the closed discussion period between reviewers + AC.

---

> > > ### Author Response · Authors · 2023-11-22
> > >
> > > Much appreciated for your reading our responses and reconsideration of the score!
> > >
> > > We understand your suggestion that beating or at least matching the strong state-of-the-art methods would make our method stronger. As you may know, all functional BNN papers we found did not compare with such a method, so we followed the experimental setup with them before. After reading the suggested paper on Laplace, we found it to be a really interesting one and different from our perspective on the same problem. We would pay more attention to this work in our further study to investigate the underlying connections between two different perspectives.

---

### Official Review · Reviewer_ExHU · 2023-10-26

**Soundness:** 3 good
**Presentation:** 3 good
**Contribution:** 3 good
**Rating:** 6
**Confidence:** 3

**Summary:**

In pursuit of assigning meaningful functional priors and securing well-behaved posteriors, this study introduces a groundbreaking approach termed "Functional Wasserstein Bridge Inference" (FWBI), diverging from traditional parameter space variational inference. The manuscript commences by critically examining the limitations inherent in the use of Kullback-Leibler (KL) divergence. Subsequently, it unveils a two-step variational inference technique, capitalizing on functional priors and a bridging distribution to directly approximate the posterior.

**Strengths:**

1. **Solid Theoretical Analysis**: The theoretical groundwork laid out in your paper is both rigorous and comprehensive. Notably, it addresses the important question of meaningful priors within the framework of variational inference, effectively advancing the understanding of this topic.

2. **Estimation of Prior**: I highly commend your innovative use of the optimal transport technique for prior estimation, a commendable departure from the more conventional approach of employing iid Gaussian distributions. This choice adds a layer of sophistication to your work and enhances its theoretical soundness.


3. **Ample Experiments**: The experimental design and execution in your manuscript are both abundant and well-executed, effectively corroborating your proposed work. The results lend considerable credibility to your methodology, reinforcing the paper’s overall impact.

**Weaknesses:**

1. **Evaluation Index**: My expertise primarily lies in regression tasks, and I wonder whether the authors have considered employing alternative evaluation metrics such as calibration curves, as suggested in Reference [1]. The inclusion of such metrics could potentially provide a more comprehensive assessment of the model's performance.

2. **Paired-sample $t$-test**: I recommend that the authors include a paired-sample $t$-test in the results section. This statistical test would serve to demonstrate the model's superiority more convincingly, and it is generally considered a robust method for comparing the means of two related groups.

3. **Definition of Convergence**: Upon reviewing Appendix D.3, I noted that the convergence line appears to be based on the learning objective. Could you please clarify your definition of convergence within this context? Furthermore, is it possible to theoretically prove the algorithm's convergence, given your chosen definition?
---
References:
[1]. Kuleshov V, et. al, Accurate uncertainties for deep learning using calibrated regression[C]//International conference on machine learning. PMLR, 2018: 2796-2804.

**Questions:**

The questions are listed in weakness. I would raise my score if the authors answer weakness 1) and  3).

---

> ### Author Response · Authors · 2023-11-22
>
> We express our gratitude to Reviewer ExHU for great suggestion on calibration cures and t-test. In response to your comments, we prepared a revised version of the manuscript and address specific questions below.
>
> > Evaluation Index: My expertise primarily lies in regression tasks, and I wonder whether the authors have considered employing alternative evaluation metrics such as calibration curves, as suggested in Reference [1]. The inclusion of such metrics could potentially provide a more comprehensive assessment of the model's performance.
>
> $\textbf{Reply:}$ As suggested, we have added the calibration curves of all parametric and functional methods for toy regression examples shown in Figure 8 in Appendix E.4, where our FWBI shows the most superior calibration results.
>
> > Paired-sample -test: I recommend that the authors include a paired-sample -test in the results section. This statistical test would serve to demonstrate the model's superiority more convincingly, and it is generally considered a robust method for comparing the means of two related groups.
>
> $\textbf{Reply:}$ We have performed the paired t test for the UCI regression results in Table 1. The p-value for any given pair (one is ours and the other is any other method) is <0.001, so we draw the conclusion that our method is significantly better than others. To save the space, we add a sentence in the caption of the table to state that “We performed the paired-sample t-test for the results from FWBI and the results from other methods and all pairs are with p < .001” instead of including all p-values in the table.
>
> > Definition of Convergence: Upon reviewing Appendix D.3, I noted that the convergence line appears to be based on the learning objective. Could you please clarify your definition of convergence within this context? Furthermore, is it possible to theoretically prove the algorithm's convergence, given your chosen definition?
>
> $\textbf{Reply:}$ The convergence figures are the convergence processes of the 1-Wasserstein distance between the bridging distribution over functions and the functional prior (the top row), and the 2-Wasserstein distance between the variational posterior and the bridging distribution (the bottom row) during the training. Additionally, we added convergence plots for the overall variational objective function of FWBI in Figures 9 and 10 of Appendix for two toy examples respectively. The convergence here means the objective function to be converge to some stable values during the optimization. The comparisons of methods are valid only when they are converged well. That is the reason why we plot them here. Note that our algorithm is to design a loss function rather than optimize the loss function. Specially, we have designed a loss function in Eq. (11) for functional BNNs problem, and then we use an optimizer (e.g., SGD or Adam) to optimize this loss function. Given a loss function, the convergence is guaranteed by the optimizer rather than our algorithm.
>
> We would again like to thank Reviewer ExHU, and we hope that our changes adequately address your concerns. Please let us know if you have any further questions or comments, and we are very happy to follow up!

---

### Official Review · Reviewer_n4ZG · 2023-10-31

**Soundness:** 4 excellent
**Presentation:** 3 good
**Contribution:** 3 good
**Rating:** 6
**Confidence:** 4

**Summary:**

The authors focus on functional priors for Bayesian neural networks, which allow the incorporation of  better-motivated priors rather than the _naive_ weight-space priors that are usually. For this task, they build an ELBO that consists of two additional Wasserstein-based loss terms regularizing the variational posterior towards the desired functional prior.

**Strengths:**

- The method is well-motivated and theoretically justified
- It is evaluated on several data sets consisting of regression and classification tasks and shows clear improvements upon its baselines
- The paper is well-written and easy to follow

**Weaknesses:**

- While the paper already evaluates several priors (RBF, Matérn, linear kernels, and a BNN), it lacks some more interesting ones, e.g., how would a periodic kernel-based prior (as mentioned in the introduction) perform in Figure 2?
- The contribution section promises "reliable uncertainty estimation", while barely providing such an experiment in the form of OOD detection. A more detailed evaluation would involve reporting predictive log-likelihoods for all regressions and classification experiments (in addition to the current RMSE/Accuracy), as well as, e.g., expected
classification errors for the classification setup
- For Figures 1/2 the mean fit looks a lot better than for the baselines, but still lacks a lot with respect to predictive uncertainty. Compare this to Figure 1 in Wild et al. (2022) who use the same function
- Sec 2, second paragraph claims "if prior and likelihood distributions are both assumed to be Gaussian, the posterior $p(\mathbf{w}|\mathcal{D})$ can be solved analytically as a Gaussian. As the posterior for neural-network-based mappings within the likelihood is highly multimodal, this is a false statement.
- Similarly, on page 3 (top) it is stated that parameter-space variational objectives can be optimized if the variational posterior "is assumed to be a fully-factorized Gaussian distribution". While true, the sentence reads as if that were the only case, but $q$ could also be a variety of other distributions and still remain tractable to stochastic gradient descent.


_Note: Especially the experimental concerns are what keeps the current score at six instead of eight which it would otherwise be in my opinion._

**Questions:**

- Figure 1/2 only visualize an FBNN baseline. What would a corresponding figure look like for your main competitor, i.e., GWI?
- How does the posterior for a GP with each of the three kernels look like for Figure 1/2?

---

> ### Author Response · Authors · 2023-11-22
>
> We express our gratitude to Reviewer n4ZG for positive and insightful feedback. In response to your comments, we prepared a revised version of the manuscript and address specific questions below.
>
> > Q1: Figure 1/2 only visualize an FBNN baseline. What would a corresponding figure look like for your main competitor, i.e., GWI?}
>
> $\textbf{Reply:}$ We have tested GWI using Figure 1’s setting and the results are given in Figure 6 (a), (b), and (c) of Appendix E.2. We can see that our FWBI still shows stronger predictive ability to recover the target function and competitive uncertainty estimation in the unseen regions. Since Figure 2 is used to show the ability of our method to use non-GP prior, we did not include them because GWI cannot use BNN non-GP priors.
>
> > Q2: How does the posterior for a GP with each of the three kernels look like for Figure 1/2?
>
> $\textbf{Reply:}$ Since we want to highlight that our method (and other functional BNNs) could easily incorporate a meaningful prior via GP contrast to parameter-space BNNs, we give a presumed meaningful GP prior to all functional BNNs in the Figures 1 and 2. To obtain a “meaningful” GP prior, we used all (both training and test) data points to train a GP. Such setting is unfair to GP. Hence, in Figure 7 of Appendix E.3, we change the prior to a GP trained only on training data points and then the results from GP (with three kernels) and our method are compared, where our method has similar large variances in the regions without data but better (smaller) variance on the regions with observed training data.
>
> > While the paper already evaluates several priors (RBF, Matérn, linear kernels, and a BNN), it lacks some more interesting ones, e.g., how would a periodic kernel-based prior (as mentioned in the introduction) perform in Figure 2?
>
> $\textbf{Reply:}$ We have added posterior results for periodic-kernel based GP prior in Figure2, and FWBI still performs well with periodic kernel compared to other methods.
>
> > The contribution section promises "reliable uncertainty estimation", while barely providing such an experiment in the form of OOD detection. A more detailed evaluation would involve reporting predictive log-likelihoods for all regressions and classification experiments (in addition to the current RMSE/Accuracy), as well as, e.g., expected classification errors for the classification setup.
>
> $\textbf{Reply:}$ We have added the NLL results for UCI regression in Table3 in Appendix E.7. For OOD detection in classification task, there are AUC results based on Entropy of the predictions shown in Table 2 in the main body, and Figure 16 in Appendix E.9 also shows the ROC curve. Our FWBI shows competitive uncertainty estimation performance compared to other methods. We also added the calibration curves of all parametric and functional methods for toy regression examples shown in Figure 8 in Appendix E.4, where our FWBI shows the most superior calibration results.
>
> > For Figures 1/2 the mean fit looks a lot better than for the baselines, but still lacks a lot with respect to predictive uncertainty. Compare this to Figure 1 in Wild et al. (2022) who use the same function.
>
> $\textbf{Reply:}$ It is possible that the experimental results will not be the same under different settings. We have added the results of Wild et al. (2022) under our settings on the toy example shown in Figure 6 of Appendix E.2, where we can see that our FWBI still shows stronger predictive ability to recover the target function and competitive uncertainty estimation in the unseen regions.
>
> > Sec 2, second paragraph claims "if prior and likelihood distributions are both assumed to be Gaussian, the posterior can be solved analytically as a Gaussian. As the posterior for neural-network-based mappings within the likelihood is highly multimodal, this is a false statement.
>
> $\textbf{Reply:}$ Sorry for the misunderstanding. In fact, our statement was intended for the general Bayesian inference not for BNNs, and we have removed this sentence in the revision to avoid the misunderstanding.
>
> > Similarly, on page 3 (top) it is stated that parameter-space variational objectives can be optimized if the variational posterior "is assumed to be a fully-factorized Gaussian distribution". While true, the sentence reads as if that were the only case, but could also be a variety of other distributions and still remain tractable to stochastic gradient descent.
>
> $\textbf{Reply:}$ Sorry for the ambiguity, and we have removed this sentence in the revision.
>
> We would again like to thank Reviewer n4ZG, and we hope that our changes adequately address your concerns. Please let us know if you have any further questions or comments, and we are very happy to follow up!

---

### Official Review · Reviewer_b5oJ · 2023-11-01

**Soundness:** 3 good
**Presentation:** 3 good
**Contribution:** 2 fair
**Rating:** 6
**Confidence:** 3

**Summary:**

The paper builds on the literature on functional variational inference for Bayesian deep learning and proposes a new Bayesian inference framework in the function space. To achieve this, the authors build a bridge process and propose to use 1-Wasserstein distance between random processes to minimize the distance between the bridge process and an apriori random process that provides a suitable prior over the functions (e.g. Gaussian Process). Then in the second step, the variational posterior over the network weights is optimized to match the bridging distribution in the parameter space where the optimization is done by minimizing the 2-Wasserstein distance between the parametric variational family and the bridging distribution. In order to avoid constraining the solution space of the bridging distribution, the authors propose to perform the first and second optimization simultaneously, and provide efficient sample-based algorithms to minimize the weighted sum of the 1-Wasserstein functional distance and 2-Wasserstein parameter distance. Experiments are done on toy datasets, contextual bandits, the UCI dataset, and image classification. In all cases, the proposed method (FWBI) outperforms compared methods in terms of accuracy. The OOD detection results are presented on the image classification dataset where FWBI shows better performance than the compared methods on two out of three datasets.

**Strengths:**

Despite their promise and excellent motivation, BDL methods have rarely been used in applied settings. This is due to a multitude of reasons. First, performing exact inference in the parameter space is intractable, leading to a handful of approximate inference techniques developed to remedy this. Second, incorporating prior knowledge about the problem space or task space in the parameter space is challenging due to a lack of clear understanding of how parameter priors translate into functional priors. This has motivated the development of building priors directly in the function space but most of the existing methods are not general enough for a wide range of applications or suffer from intractability of inference in high dimensions. This paper proposes an alternative framework based on Wasserstein bridges that remedies the existing issues and appears to possess strong prediction and calibration properties.

The paper is self-contained and the presentation of arguments follows a logical order.

Including both quantitative and qualitative results on multiple datasets is an important strength of the paper.

**Weaknesses:**

Some of the strong existing methods (see the questions section) are not included in the comparisons. I believe even if the existing methods outperform FWBI in some datasets this paper is still a valuable contribution. So including those results will just add to the strength of the paper.

Although there's a contribution section with a list of proposed contributions, I still had a hard time disentangling what existed in the literature and what was new in the paper (see the questions section).

Although BDL methods originally were developed for calibration purposes, I don't see many calibration results in the paper. It's a common practice in BDL papers to present results on IND and OOD calibration on real and toy datasets. Again, this will add to the strength of the paper and help practitioners assess the circumstances under which FWBI achieves its best performance.

Results are not systematically shown as a function of varying the dimension, noise, and smoothness of the prior, among other factors.

**Questions:**

Parts of the presented methods are very similar to [1]. Although the authors cite the paper in the paragraph where their method is used the similarities and differences between the two methods are not clearly stated. In addition, despite sharing a lot of similarities that paper is not included in the compared methods among the functional prior BNNs. Please include a more detailed discussion of what exactly are the differences and how the methods compare in terms of their performance.

Although the main motivation of the presented methods is uncertainty estimation and calibration most of the focus of the paper is on the in-distribution accuracy measurements as opposed to calibration properties. Common ways of measuring calibration are to use an in-distribution (IND) test dataset (such as MNIST) and an out-of-distribution (OOD) test dataset (such as Not-MNIST) and evaluate metrics such as ECE, MCE, Entropy of the predictions, calibrated, etc. Can the authors include these results to show that the model class proposed here achieves better calibration?

In Fig. 2, why does the model provide such narrow uncertainty bounds for regions without data? This is a behavior that’s not expected from a well-calibrated model such as GP with RBF kernel.

I’m not quite sure how to interpret the results presented in Figs. 1, 2. Do the error bars represent standard deviation? If so, to me a method that does a reasonable job of calibration should have error bars that cover the true function. This seems to be the case only for the KLBBB method. How do the authors justify these results?

Can the authors include GP as the gold standard for the comparisons in FIg. 1, 2? Also, can you include an MCMC method such as HMC performed on BNN with normal weights (use a small BNN so that HMC can run in a reasonable amount of time, or use SG-MCMC for intermediate BNNs) in compared methods?

A recent work [2] proposes to use Lipschitz functions in the architecture of a network to achieve what they call “distance awareness” which will lead to proper uncertainty estimation. While the use of Lipschitz functions in that work is motivated from a completely different perspective it seems to be a shared component of your work (and the original work by [1]).

[1] https://arxiv.org/abs/2011.12829

[2] https://arxiv.org/abs/2205.00403

---

> ### Author Response · Authors · 2023-11-22
> **Response part one (for question 1-5) for reviewer b5oJ**
>
> We express our gratitude to Reviewer b5oj for positive and insightful feedback. In response to your comments, we prepared a revised version of the manuscript and address specific questions below.
>
> > Q1: Parts of the presented methods are very similar to [1]. Please include a more detailed discussion of what exactly are the differences and how the methods compare in terms of their performance.
>
> $\textbf{Reply:}$ Firstly, we want to highlight that [1] is not for functional variational inference. The task of [1] is the prior matching. The different tasks are the first major difference between ours with [1], and that is the reason why we did not include it as comparative method. The other difference is that the 1-Wasserstein distance to measure the distance between two stochastic processes is only one terms of our algorithm, and we have another two terms: one is log-likelihood and the other is the distance between variational posterior and bridging distribution. The three terms are jointly optimized in a functional variational inference framework which is our main contribution in this study.
>
> > Q2:  Can the authors include these results to show that the model class proposed here achieves better calibration?
>
> $\textbf{Reply:}$ We have added the NLL results for UCI regression in Table3 in Appendix E.7. For OOD detection in classification task, there are AUC results based on Entropy of the predictions shown in Table 2 in the main body, and Figure 16 in Appendix E.9 also shows the ROC curve. Our FWBI shows competitive uncertainty estimation performance compared to other methods. We also added the calibration curves of all parametric and functional methods for toy regression examples shown in Figure 8 in Appendix E.4, where our FWBI shows the most superior calibration results.
>
> > Q3: In Fig. 2, why does the model provide such narrow uncertainty bounds for regions without data? This is a behavior that’s not expected from a well-calibrated model such as GP with RBF kernel.
>
> $\textbf{Reply:}$ The reason why our method looks like producing over-confident predictions is because we used a “meaningful” GP as prior. Since we want to highlight that our method (and other functional BNNs) could easily incorporate a meaningful prior via GP contrast to parameter-space BNNs, we give a presumed meaningful GP prior to all functional BNNs in the Figures 1 and 2. To obtain a “meaningful” GP prior, we used all (both training and test) data points to train a GP. Such prior gives FBNNs stronger predictive capabilities in the area even without training data, which makes our method look like over-confident. In Figure 7 of Appendix E.3, we change the prior to a GP trained only on training data points, and then our method would not look like over-confident any more as demonstrated by the large variances in the areas without training data.
>
> > Q4: I’m not quite sure how to interpret the results presented in Figs. 1, 2. Do the error bars represent standard deviation? If so, to me a method that does a reasonable job of calibration should have error bars that cover the true function. This seems to be the case only for the KLBBB method. How do the authors justify these results?
>
> $\textbf{Reply:}$ Yes, you are right that the error bars represent the standard deviation. Note that the error bars from a reasonable method should cover the true function only in the regions with training data. For the regions without training data, there is no guarantee for the coverage. The best method should be able to cover the true function in the regions with training data with small standard deviation and show relatively larger standard deviation in the regions without training data.
>
> > Q5: Can the authors include GP as the gold standard for the comparisons in FIg. 1, 2? Also, can you include an MCMC method such as HMC performed on BNN with normal weights (use a small BNN so that HMC can run in a reasonable amount of time, or use SG-MCMC for intermediate BNNs) in compared methods?
>
> $\textbf{Reply:}$ The figure 2 is used to show the ability of our method to use non-GP prior, so we only added SG-MCMC results using Figure 1’s setting and the results are shown in Figure 6(d) of Appendix. Since we used a meaningful GP prior for our method in Figure 1, the setting would be unfair for GP. Hence, we also compare GP with our methods in In Figure 7 of Appendix E.3, we change the prior to a GP trained only on training data points, and then our method would not look like over-confident any more as demonstrated by the large variances in the areas without training data.

---

> ### Author Response · Authors · 2023-11-22
> **Response part two for Reviewer b5oJ**
>
> > Results are not systematically shown as a function of varying the dimension, noise, and smoothness of the prior, among other factors.
>
> $\textbf{Reply:}$ We have added analysis for the impact of functional properties (smoothness and noise) of prior on FWBI posterior in Appendix E.6. Consider a 1-D periodic function: $y = 2 * \sin(4x) + \epsilon$ with noise $\epsilon \sim \mathcal{N}(0,0.01)$, and randomly sample 20 training points from this function within $[-2, -0.5] \cup[0.5, 2]$. Firstly, we fit a GP with Matern kernel as the prior using these training data. Since the Matern kernel has a parameter $\nu$ used to control the smoothness of the functions from GP, we use $\nu$ to simulate different prior smoothness. The results are shown in Figure 12 of Appendix E.6, where Figures 12(a) and 12(b) are the results from the unsmoothed ($\nu=0.5$) and smoothed ($\nu=2.5$) GP priors, respectively. The corresponding FWBI posteriors are shown in Figures 12(c) and 12(d), where we can see the smoothness of prior has very limited effects on the resulting posteriors.
>
> Then, we investigate the impact of prior noise on the posteriors by adding different GP noises to the pre-trained GP prior (with periodic kernel). Specifically, we consider two situations: GP noises with fixed 0 mean and varying variances, and GP noises with fixed variance and varying means, respectively. The results are shown in Figures 13 and 14 of Appendix E.6, where the left column shows several GP priors with different injected noises, and the right column shows the corresponding FWBI posteriors. We can observe that: 1) when the mean of noises is fixed, there is no significant effect from varying variances (from 0.5 to 5) on the region with training data, and only a minor effect on the right-hand side region without observed training data. The larger variances tend to destroy the prediction on the non-data regions; 2) when the variance of noises is fixed, varying means (from 0.3 to 3) also have little effect on the region with training data but destroy the prediction on the non-data regions. The larger changed means would lead to worse prediction.
>
> We would again like to thank Reviewer b5oj, and we hope that our changes adequately address your concerns. Please let us know if you have any further questions or comments, and we are very happy to follow up!

---

> ### Author Response · Authors · 2023-11-22
> **Response to question 6 for Reviewer b5oJ**
>
> > A recent work [2] proposes to use Lipschitz functions in the architecture of a network to achieve what they call “distance awareness” which will lead to proper uncertainty estimation. While the use of Lipschitz functions in that work is motivated from a completely different perspective it seems to be a shared component of your work (and the original work by [1]).
>
> $\textbf{Reply:}$ Different from our main BDL topic, work[2] studied approaches to improve the uncertainty property of a single DNN.  They uses the bi-Lipschitz condition to make the hidden mapping distance preserving and further improve the uncertainty quality of a single DNN. Specifically, the upper Lipschitz bound prevents the hidden representations from overly sensitive to the meaningfulness perturbations, and the lower Lipschitz bound prevents the hidden representation from collapsing the representations of distinct examples together. While our work is motivated by different perspectives, in which the Lipschitz function is used only for the estimation of the 1-Wasserstein distance between the functional prior and the bridging distribution.

---

### Official Review · Reviewer_RTAX · 2023-11-06

**Soundness:** 2 fair
**Presentation:** 3 good
**Contribution:** 3 good
**Rating:** 3
**Confidence:** 3

**Summary:**

The paper focuses on function-space inference for Bayesian deep learning models and proposes a novel variational method termed functional Wasserstein bridge inference (FWBI). Typical function-space VI approaches define the similarity between distributions over functions in terms of the Kullback-Leibler divergence, but this measure is known to be ill-defined for such distributions. As an alternative similarity measure, the authors propose to use the Wasserstein distance between the variational posterior and the (parametric) prior, which itself is fitted to a functional prior (e.g., a Gaussian process) by minimising another Wasserstein distance. The resulting objective contains the likelihood and the two Wasserstein distances, but the authors show (in the appendix) that this is a proper lower bound on the model evidence. Finally, they compare FWBI against both weight-space and function-space inference approaches, showing good predictive performance and uncertainty quantification.

**Strengths:**

1. The paper presents an interesting idea that addresses an important issue. It is clearly interesting to the ICLR community and is likely to become a significant reference in the field of function-space inference.
2. The work appears to be original, although it seems to be a combination of the methods presented by Tran et al. (2020) and Wild et al. (2022). The authors are entirely open about this and cite both papers.
3. The paper is clearly written and seems to be of high technical quality.
4. Empirically, the method seems to work well and has a very competitive running time.


References:
Tran et al., Functional Priors for Bayesian Neural Networks through Wasserstein Distance Minimization to Gaussian Processes, AABI 2020.
Wild et al., Generalized Variational Inference in Function Spaces: Gaussian Measures meet Bayesian Deep Learning, NeurIPS 2022.

**Weaknesses:**

1. I am mainly missing some empirical analysis of the proposed objective compared to competitors. The running time is faster, but how does, say, the stability of the training or the convergence speed compare to other methods?
2. The proposed objective function contains two hyperparameters weighing the contributions from each of the Wasserstein distances, which makes sense practically, although it is a little unsatisfying theoretically. The authors do not discuss how to choose or tune these hyperparameters.
3. The experiments use the functional BNNs of Sun et al. (2019) as the main competitor, but more recent and stronger baselines exist, for instance, Ma & Hernández-Lobato (2021) and Rudner et al. (2022), which are both cited by the authors. It would have been fair to at least include one such baseline in the experimental evaluation.
4. The authors highlight uncertainty quantification as one of the method's strengths, but no uncertainty quantification metrics (say, predictive log-likelihood) are provided for the UCI experiments.
5. It is a shame that quite many important results, such as the proof that the objective is an ELBO and some of the empirical analysis, have been put in the appendix. There is, of course, a limit to how much one can fit on 9 pages, but for instance the introduction and preliminaries could perhaps have been shortened a bit to at least make room for a proof sketch. Given that this proof is highlighted as a contribution, at least something should be said about it in the main paper.

References:
Ma & Hernández-Lobato, Functional Variational Inference based on Stochastic Process Generators, NeurIPS 2021.
Rudner et al. Tractable Function-Space Variational Inference in Bayesian Neural Networks, NeurIPS 2022.

**Questions:**

1. In section 3.2, you call $\boldsymbol{\theta_q}$ and $\boldsymbol{\theta_q}$ "stochastic parameters". It also seems like they are sampled in Eq. (12), but in Eq. (11), they are minimised, which confuses me. Can you elaborate on this?
2. Did you compute uncertainty quantification metrics, such as the predictive log-likelihood, for the UCI experiments? Based on figures 1 and 2, it seems to me that FWBI produces quite over-confident predictions.
3. Do you have advice on how to select or tune the hyperparameters in the objective?
4. What does the running time in E.2 measure? Is it training time or prediction time? If training time, is it per epoch or until convergence?

---

> ### Author Response · Authors · 2023-11-22
>
> We express our gratitude to Reviewer RTAX for the support. In response to your comments, we prepared a revised version of the manuscript and address specific questions below. **
>
> > Q1: In section 3.2, you call  $\theta_q$ and  $\theta_q$ "stochastic parameters". It also seems like they are sampled in Eq. (12), but in Eq. (11), they are minimised, which confuses me. Can you elaborate on this?
>
> $\textbf{Reply:}$ Sorry for the confusion! $\theta_q$ is the stochastic variational parameter of the optimization problem in Eq. (11). Once the optimal $\theta_q^*: =(\mathbf{\mu}_q^*, \mathbf{\Sigma}_q^*)$ is obtained, we can obtain the approximation posterior predictive function draws by sampling weights $\mathbf{w}^{(j)}$ from $\mathcal{N}(\mathbf{\mu}_q^*, \mathbf{\Sigma}_q^*)$. We have revised the Eq. (12) as follow:
>
> $q(\mathbf{y}^*|\mathbf{x}^*) \approx \frac{1}{S} \sum_{j=1}^{S} p(\mathbf{y}^*|f(\mathbf{x}^*; \mathbf{w}^{(j)})), \mathbf{w}^{(j)} \sim \mathcal{N}(\mathbf{\mu}_q^*, \mathbf{\Sigma}_q^*), \boldsymbol{\theta_q}^* := (\mathbf{\mu}_q^*, \mathbf{\Sigma}_q^*).$
>
> > Q2: Did you compute uncertainty quantification metrics, such as the predictive log-likelihood, for the UCI experiments? Based on figures 1 and 2, it seems to me that FWBI produces quite over-confident predictions.
>
> $\textbf{Reply:}$ We have added the NLL results for UCI regression in Table3 in Appendix E.7. Our FWBI shows competitive uncertainty estimation performance compared to other methods. Furthermore, we also added the calibration curves as recommended by a reviewer and the results are shown in Figure 3 in Appendix E.4, where our methods have shown better performances as well.
>
> The reason why our method looks like producing over-confident predictions is because we used a “meaningful” GP as prior. Since we want to highlight that our method (and other functional BNNs) could easily incorporate a meaningful prior via GP contrast to parameter-space BNNs, we give a presumed meaningful GP prior to all functional BNNs in the Figures 1 and 2. To obtain a “meaningful” GP prior, we used all (both training and test) data points to train a GP. Such prior gives FBNNs stronger predictive capabilities in the area even without training data, which makes our method look like over-confident. In Figure 7 of Appendix E.3, we change the prior to a GP trained only on training data points, and then our method would not look like over-confident any more as demonstrated by the large variances in the areas without training data.
>
> > Q3: Do you have advice on how to select or tune the hyperparameters in the objective?
>
> $\textbf{Reply:}$ We did not use any hyperparameter tuning method to obtain the optimal ones but only chose the hyperparameters to ensure different components are within the same scale. We believe our method could achieve further better performance after thorough hyperparameter tuning using some black-box optimization methods, like Optuna and scikit-optimize.
>
> > Q4: What does the running time in E.2 measure? Is it training time or prediction time? If training time, is it per epoch or until convergence?
>
> $\textbf{Reply:}$ The running time in E.2 is the total training time for all 2000 training epochs.
>
> > I am mainly missing some empirical analysis of the proposed objective compared to competitors. The running time is faster, but how does, say, the stability of the training or the convergence speed compare to other methods?
>
> $\textbf{Reply:}$ We have added a plot of convergence processes of training loss for our FWBI in Figures 9 and 10 in Appendix E.5 on toy examples and convergence processes for all methods in Figure 15 in Appendix E.8 on UCI regressions. Our FWBI demonstrates better convergence speed and stability compared to other methods.
>
> > The experiments use the functional BNNs of Sun et al. (2019) as the main competitor, but more recent and stronger baselines exist, for instance, Ma and Hernández-Lobato (2021) and Rudner et al. (2022), which are both cited by the authors. It would have been fair to at least include one such baseline in the experimental evaluation.
>
> $\textbf{Reply:}$ We did try to include them as our competitive methods. However, they did not publish their codes, and we tried to ask for the code from the authors by email, but unfortunately, didn't get any reply.
>
> > Given that this proof is highlighted as a contribution, at least something should be said about it in the main paper.
>
> $\textbf{Reply:}$ Thanks for the suggestion! We did move a lot of content to the Appendix due to the page limitation. As recommended by the reviewer, we move the statement of the theorem to the main body but have to still leave the proof details in Appendix.
>
> We would again like to thank Reviewer RTAX, and we hope that our changes adequately address your concerns. Please let us know if you have any further questions or comments, and we are very happy to follow up!

---

### Author Response · Authors · 2023-11-22
**A revised version of the manuscript has been uploaded**

We express our gratitude to all reviewers for your valuable comments. In response to your comments, we updated a revised version of the manuscript and addressed specific questions. In order to distinguish it from the previous version, we have highlighted the changes in the main text.

---

### Meta-Review · Area_Chair_vVyR · 2023-12-06

**Metareview:**

This paper proposes a novel variational inference scheme for BNNs in function space, where the ill-defined KL divergence is replaced with a Wasserstein bridge to allow the use of function-space priors and yield tractable function-space posteriors. After an active discussion between authors and reviewers, the paper is very much borderline, with three reviewers leaning accept and two reject. While the reviewers praised the importance of the problem and strength of the theory, they were critical of the novelty of the contributions, weak empirical results, lack of ablation studies, and lack of baselines. Unfortunately, the criticism seems to outweigh the praise at the time being. I believe that this could be a really interesting contribution and would encourage the authors to take the reviewer feedback seriously and resubmit a revised version of the paper in the future.

**Justification For Why Not Higher Score:**

see points of criticism above

**Justification For Why Not Lower Score:**

N/A

---

### Decision · Program_Chairs · 2024-01-16

Reject